# Global aviation contrail climate effects from 2019 to 2021

**Roger Teoh[1], Zebediah Engberg[2], Ulrich Schumann[3], Christiane Voigt[3,4], Marc Shapiro[2], Susanne Rohs[5], and Marc E. J. Stettler[1]**

[1]Department of Civil and Environmental Engineering, Imperial College London, London, SW7 2AZ, United Kingdom
[2]Breakthrough Energy, 4110 Carillon Point, Kirkland, WA 98033, United States
[3]Institute of Atmospheric Physics, Deutsches Zentrum für Luft- und Raumfahrt, 82234 Oberpfaffenhofen, Germany
[4]Institute of Atmospheric Physics, University Mainz, 55099 Mainz, Germany
[5]IEK-8 Troposphäre, Institut für Energie und Klimaforschung, Forschungszentrum Jülich GmbH, Jülich, Germany

**Correspondence:** Marc E. J. Stettler (m.stettler@imperial.ac.uk)

**Abstract.** The current best-estimate of the global annual mean radiative forcing (RF) attributable to contrail cirrus is thought to be 3 times larger than the RF from aviation's cumulative $CO_2$ emissions. Here, we simulate the global contrail RF for 2019–2021 using reanalysis weather data and improved engine emission estimates along actual flight trajectories derived from Automatic Dependent Surveillance–Broadcast telemetry. Our 2019 global annual mean contrail net RF ($62.1\,\mathrm{mW\,m^{-2}}$) is 44 % lower than current best estimates for 2018 (111 [33, 189] $\mathrm{mW\,m^{-2}}$, 95 % confidence interval). Regionally, the contrail net RF is largest over Europe ($876\,\mathrm{mW\,m^{-2}}$) and the USA ($414\,\mathrm{mW\,m^{-2}}$), while the RF values over East Asia ($64\,\mathrm{mW\,m^{-2}}$) and China ($62\,\mathrm{mW\,m^{-2}}$) are close to the global average, because fewer flights in these regions form persistent contrails resulting from lower cruise altitudes and limited ice supersaturated regions in the subtropics due to the Hadley Circulation. Globally, COVID-19 reduced the flight distance flown and contrail net RF in 2020 ($-43$ % and $-56$ %, respectively, relative to 2019) and 2021 ($-31$ % and $-49$ %, respectively) with significant regional variations. Around 14 % of all flights in 2019 formed a contrail with a net warming effect, yet only 2 % of all flights caused 80 % of the annual contrail energy forcing. The spatiotemporal patterns of the most strongly warming and cooling contrail segments can be attributed to flight scheduling, engine particle number emissions, tropopause height, and background radiation fields. Our contrail RF estimates are most sensitive to corrections applied to the global humidity fields, followed by assumptions on the engine particle number emissions, and are least sensitive to radiative heating effects on the contrail plume and contrail–contrail overlapping. Using this sensitivity analysis, we estimate that the 2019 global contrail net RF could range between 34.8 and $74.8\,\mathrm{mW\,m^{-2}}$.

## 1 Introduction

Aviation contributes to significant social and economic benefits, but it also emits $CO_2$ and non-$CO_2$ pollutants that cause global warming and degrade air quality. In particular, aviation's cumulative $CO_2$ emissions account for one-third of its overall effective radiative forcing (ERF), while the remaining two-thirds are estimated to arise from non-$CO_2$ components such as contrail cirrus, nitrogen oxides ($NO_x$), particulate matter, and stratospheric water vapour emissions (Lee et al., 2021). When taken together, aviation was responsible for $\sim 3.5$ % of the global anthropogenic radiative forcing (RF) in 2018 with contrail cirrus estimated to contribute more than half of the aviation-induced RF (Lee et al., 2021; Kärcher, 2018).

Contrails form when flights traverse air masses with ambient temperatures below the Schmidt–Appleman criterion (SAC) threshold temperature ($T_{SAC}$, typically $< 230$ K). Aircraft particle emissions, which consist of non-volatile particulate matter (nvPM), metallic compounds, semi-volatile organic and sulfuric particles (Petzold et al., 2005), and ambient natural aerosols can activate into droplets and freeze to form contrail ice crystals (Schumann, 1996; Kärcher, 2018; Kleine et al., 2018). The nvPM acts as the primary source of condensation nuclei in the "soot-rich" regime, defined when the soot number emissions index ($EI_n$) exceeds a threshold of around $10^{14}$ kg$^{-1}$, while ambient aerosols and organic and sulfuric particles can nucleate under "soot-poor" conditions ($EI_n < 10^{14}$ kg$^{-1}$) (Kärcher and Yu, 2009; Kärcher, 2018). Most kerosene-burning aircraft engines typically have nvPM $EI_n$ of $10^{14}$–$10^{16}$ kg$^{-1}$ (EASA, 2021; Petzold et al., 1999; Moore et al., 2017; Durdina et al., 2017), and for these aircraft types, in situ measurements and modelling studies show that the nvPM $EI_n$ influences various contrail properties and associated climate forcing (Voigt et al., 2021; Bräuer et al., 2021b; Teoh et al., 2022; Jeßberger et al., 2013; Kärcher, 2016). However, there is a small but increasing share of aircraft types powered by staged combustors with nvPM $EI_n$ as low as $\sim 10^{11}$ kg$^{-1}$ (EASA, 2021; Boies et al., 2015) for which the initial contrail properties need further investigation (Voigt et al., 2022).

Over time, contrails formed in ice supersaturated regions (ISSRs) can persist, spread, and mix with natural cirrus and transition into CE1 contrail cirrus clusters with observed lifetimes of up to 19 h (Haywood et al., 2009). The contrail spreading rate and coverage area is predominantly determined by horizontal wind components, wind shear, and ice crystal sedimentation, while contrail lifetime is dependent on the ambient relative humidity with respect to ice (RHi), atmospheric turbulence, and the rate of ice crystal losses (Schumann and Heymsfield, 2017; Lewellen et al., 2014; Lewellen, 2014; Li et al., 2023). Contrail cirrus also interacts with solar and terrestrial radiation in two distinct ways. Firstly, it reflects incoming shortwave (SW) radiation, contributing to a cooling effect during the day, while trapping and re-emitting outgoing longwave (LW) radiation, causing a warming effect at all times (Meerkötter et al., 1999). Secondly, the absorption of SW and LW radiation heats up the contrail, which can drive plume-internal turbulence and local updraughts, thereby changing the plume RHi and sublimation rate of contrail ice crystals (Jensen et al., 1998; Schumann et al., 2010; Schumann and Heymsfield, 2017; Lewellen, 2014; Unterstrasser and Gierens, 2010).

Previous studies have utilised air traffic data for 2002 (Eyers et al., 2005) and 2006 (Wilkerson et al., 2010) to estimate the global annual mean contrail cirrus net RF. Using the European Centre/Hamburg general circulation model version 4 (ECHAM4), Burkhardt and Kärcher (2011) estimated the 2002 global annual mean contrail net RF to be 37.5 mW m$^{-2}$. Bock and Burkhardt (2016a) updated the representation of contrail microphysical and optical properties within the ECHAM5 climate model (Lohmann et al., 2008; Bock and Burkhardt, 2016b) and estimated the global contrail net RF for 2002 (35 mW m$^{-2}$) and 2006 (56 mW m$^{-2}$). A follow-up study lowered the 2006 estimate by 22 % (from 56 to 44 mW m$^{-2}$), because the initial contrail ice crystal numbers from Bock and Burkhardt (2016a) did not account for (i) the lower nvPM activation rate when ambient temperatures are close to $T_{SAC}$ and (ii) ice crystal losses in the wake vortex phase (Bier and Burkhardt, 2022). Chen and Gettelman (2013) applied the Community Atmosphere Model (CAM5) to obtain a 2006 global contrail net RF of $13 \pm 10$ mW m$^{-2}$, but this RF value was later revised to 57 mW m$^{-2}$ after the simulation was re-run with initial contrail properties that are consistent with in situ measurements (Lee et al., 2021). Schumann et al. (2015) coupled the contrail cirrus prediction (CoCiP) model with CAM3 to account for humidity exchange between contrails and the background air, estimating the 2006 global contrail net RF to be 63 mW m$^{-2}$ (or 74 mW m$^{-2}$ without humidity exchange).

Lee et al. (2021) compiled results from these studies and used the growth in annual flight distance flown to extrapolate the 2006 global annual mean contrail net RF to 2018 (111 [33, 189] mW m$^{-2}$, 95 % confidence interval). However, the extrapolation could lead to inaccuracies because CE2 (i) the formation and climate forcing of contrails has a spatiotemporal dependence (Lamquin et al., 2012; Schumann et al., 2012; Bier and Burkhardt, 2022), (ii) air traffic growth was not uniform across the globe (ICAO, 2014, 2016), and (iii) there can be significant inter-annual variability in the contrail climate forcing (Wilhelm et al., 2021; Teoh et al., 2022). In addition, existing global contrail studies generally assume constant particle number emissions which do not account for differences in nvPM $EI_n$ between aircraft–engine types; therefore, they were unable to evaluate the variabilities in the contrail climate forcing that arise from individual flights and identify the set of flights with strongly warming/-cooling contrails. Regional studies have also found that 80 % of the contrail climate forcing was caused by 2 %–12 % of all flights (Teoh et al., 2020, 2022), but the applicability of these findings on the global level remain unknown.

In this study, we use a new Global Aviation emissions Inventory based on Automatic Dependent Surveillance–Broadcast (ADS-B) telemetry (GAIA) (Teoh et al., 2024) to (i) quantify the global contrail properties and climate forcing for 2019–2021 (Sect. 3.1); (ii) identify the set of conditions that causes flights to form strongly warming/cooling contrails (Sect. 3.1.4); (iii) evaluate the sensitivity of the simulated contrail climate forcing to aircraft emissions, meteorology, and contrail model parameters (Sect. 3.2); and (iv) compare our global contrail RF estimates with existing studies (Sect. 3.3).

## 2  Materials and methods

This section outlines the datasets and models used to achieve the stated research objectives. Section 2.1 describes a new global aviation emissions inventory, which includes actual trajectories, aircraft performance parameters, and nvPM emissions from individual flights (Teoh et al., 2024). Section 2.2 provides an overview of the European Centre for Medium-Range Weather Forecasts (ECMWF) ERA5 high-resolution realisation (HRES) reanalysis dataset (ECMWF; 2021; Hersbach et al., 2020) and extends an existing humidity correction model to address known limitations in the ERA5 HRES humidity fields. Section 2.3 describes the CoCiP model (Schumann, 2012; Schumann et al., 2012); Sect. 2.4 summarises the climate forcing metrics used in this study, while Sect. 2.5 sets out the difference in model set-up that is used to conduct a sensitivity analysis. Figure 1 summarises the datasets, models, and input parameters that are used in this study. Further methodological information on the (i) formulation of the extended humidity correction model, (ii) various output formats provided by CoCiP, and (iii) approach to simulate the effects of contrail–contrail overlapping is described in detail in the Supplement.

### 2.1  Global aviation emissions inventory

Global airspace surveillance systems have been transitioning towards the ADS-B standard, which enables real-time tracking of flights at high spatiotemporal resolutions and over remote regions that previously lack radar coverage (ICAO, 2021; EUROCONTROL, 2021). Recently, Teoh et al. (2024) used global ADS-B telemetry data to derive historical flight trajectories and develop a new aviation emissions inventory for 2019–2021. The dataset, known as the Global Aviation emissions Inventory based on ADS-B (GAIA), captures 103.7 million unique flights and contains the: (i) flight metadata, including the unique flight identifier, origin and destination airports, and aircraft–engine type, and (ii) flight-waypoint data provided at time intervals of 40–60 s, including the 3D position, time, fuel consumption, aircraft mass, overall efficiency, and the nvPM $EI_n$. All flights are assumed to be powered by conventional Jet A-1 fuel.

Individual aircraft types can be powered by different engine options (Quadros et al., 2022; Teoh et al., 2024), where their nvPM $EI_n$ can vary by up to 5 orders of magnitude and influence various contrail properties (Schumann, 1996; EASA, 2021; Teoh et al., 2022). However, due to the lack of data, previous contrail studies assigned a default aircraft–engine combination that is provided by the Base of Aircraft Data (BADA) aircraft performance model (Teoh et al., 2020; Schumann et al., 2021; Teoh et al., 2022). GAIA partially addresses this limitation by using the registered aircraft tail number to extract the specific aircraft variant and engine model from a global fleet database (Cirium, 2022) whenever possible, covering 59 % of all flights or 79 % of flights with jet aircraft (Teoh et al., 2024).

For each flight, GAIA uses (i) BADA 4.2 and 3.15 to estimate the fuel consumption and overall efficiency (EUROCONTROL, 2016, 2019); (ii) regional monthly passenger load factors to estimate the fuel requirements and initial aircraft mass; (iii) the ICAO Aircraft Engine Emissions Databank (EDB) to construct the nvPM emissions profile for each engine type (EASA, 2021); and three approaches to estimate the nvPM $EI_n$ including (iv) the $T_4/T_2$ methodology (Teoh et al., 2022, 2024), which uses the ratio of turbine-inlet temperature ($T_4$) to compressor-inlet temperature ($T_2$) to interpolate the nvPM emissions profile provided by the ICAO EDB (82 % of the total flight distance flown); (v) the fractal aggregates model (Teoh et al., 2019, 2020) for older engine types without engine-specific nvPM measurements (11 % of flight distance flown); and (vi) a constant value of $10^{15}\,\mathrm{kg}^{-1}$ for remaining flights without engine-specific data (Teoh et al., 2020; Schumann et al., 2015). Further information on GAIA is detailed in Teoh et al. (2024).

### 2.2  Meteorology

Global meteorological and radiation data are provided by the ERA5 HRES reanalysis (Hersbach et al., 2020), which can be publicly downloaded from the ECMWF Copernicus Climate Data Store at a longitude–latitude grid resolution of $0.25° \times 0.25°$ over 37 pressure levels and at a 1 h time resolution (ECMWF, 2021). At altitudes of 25 000–42 000 ft (7620–12 800 m) TS1, we note that the ERA5 HRES reanalysis provides meteorological data at six pressure levels which corresponds to a mean vertical resolution of $\sim$ 3000 ft (914 m).

The simulated contrail properties and lifetimes are highly sensitive to RHi (Schumann, 2012; Schumann et al., 2021; Teoh et al., 2022). However, humidity fields provided by ERA5 products contain several limitations. First, the ERA5-derived ISSR coverage area could be overestimated relative to radiosonde measurements (Agarwal et al., 2022) or underestimated when compared with in situ measurements (Reutter et al., 2020). Second, the RHi magnitude within ISSRs are generally weakly supersaturated (RHi $\approx$ 100 %), rarely exceeding RHi > 120 %, and are inconsistent with in situ measurements (Reutter et al., 2020; Gierens et al., 2020; Teoh et al., 2022). Teoh et al. (2022) recently developed a humidity correction methodology so that the corrected RHi from the ERA5 HRES has a probability density function that is consistent with in situ measurements from the European research infrastructure In-service Aircraft for a Global Observing System (IAGOS) (Petzold et al., 2015; Boulanger et al., 2022),

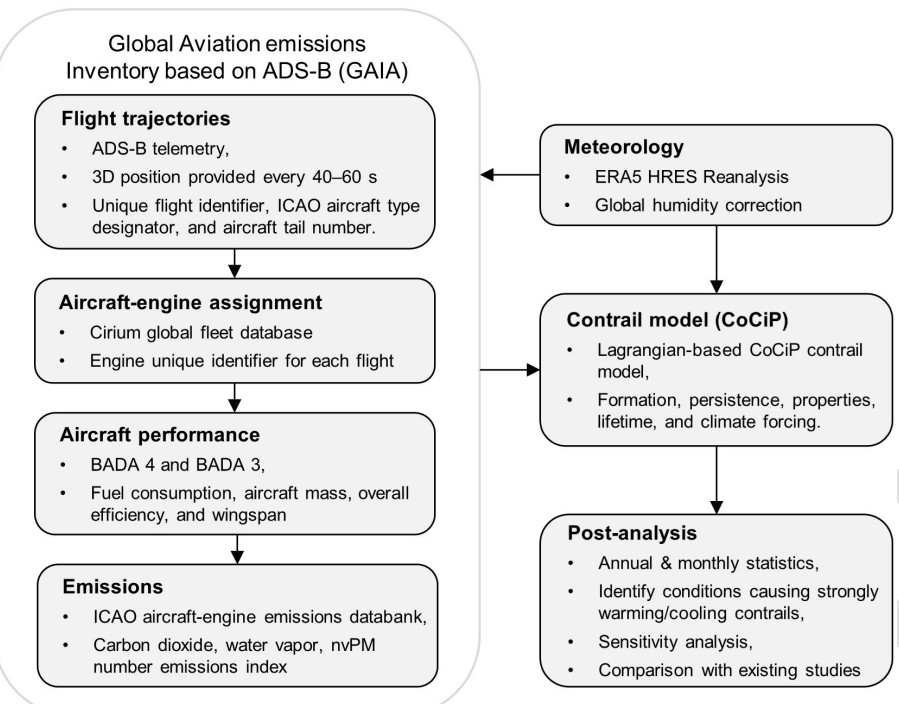

**Figure 1.** Flowchart summarising the dataset, models, and input parameters that are used in this study.

50° N/S (Santer et al., 2003),

$$RHi_{corrected} = \begin{cases} \dfrac{RHi}{a_{opt}} & \text{for } \left(\dfrac{RHi}{a_{opt}}\right) \le 1, \\ \min\left(\left(\dfrac{RHi}{a_{opt}}\right)^{b_{opt}}, RHi_{max}\right), \\ & \text{for } \left(\dfrac{RHi}{a_{opt}}\right) > 1, \end{cases} \quad (1)$$

where $a_{opt} = 0.9779$, $b_{opt} = 1.635$, and $RHi_{max} = 1.65$ were calibrated for the North Atlantic region (40–75° N and 10–50° W). However, these coefficients cannot be applied glob-
5 ally, because the RHi errors have a latitude dependence where the ERA5-derived ISSR coverage area could be over-predicted at the tropics and subtropics (0–40° N) and under-predicted at latitudes above 40° N (Table S1 in the Supplement).
10    To simulate contrails globally, we extend Eq. (1) by using the global IAGOS dataset for 2019 (Petzold et al., 2020; Boulanger et al., 2022), consisting of 2161 flights and 682 308 data points. The IAGOS dataset is split into latitude bands of 10° intervals to avoid oversampling at specific lat-
15 itudes. For each latitude band, $a_{opt}$ is optimised so that the ISSR occurrence from the ERA5 and IAGOS have a symmetrical false positive and negative rate, while $b_{opt}$ is optimised by minimising the Cramér–von Mises test statistic (Parr and Schucany, 1980) so that the RHi distribution is consistent
20 with in situ measurements (Table S2). The optimised $a_{opt}$ and $b_{opt}$ for each latitude band are then fitted with a sigmoid to capture the rapid change in tropopause height between 20–

$$a_{opt} = \frac{0.06262}{1 + \exp(0.4589 \times (|lat| - 39.25))} + 0.9522, \quad (2)$$

$$b_{opt} = \frac{1.471}{1 + \exp(0.04431 \times (|lat| - 18.76))} + 1.433. \quad (3)$$   25

We also revise $RHi_{max}$ to ensure that $RHi_{corrected}$ is thermodynamically realistic, i.e. below water saturation and below the threshold that leads to homogeneous ice nucleation (Pruppacher et al., 2007; Kärcher and Lohmann, 2002; Tompkins et al., 2007),   30

$$RHi_{max} = \begin{cases} \dfrac{p_{liq}(T_w)}{p_{ice}(T_w)}, & \text{when } T_w > 235\,K; \\ 1.67 + (1.45 - 1.67) \times \dfrac{(T_w - 190)}{(235 - 190)}, \\ & \text{when } T_w \le 235\,K; \end{cases} \quad (4)$$

where $T_w$ is the ambient temperature, and $p_{liq}(T_w)$ and $p_{ice}(T_w)$ are the saturation pressures of water vapour over liquid water and ice, respectively, that are estimated using Eqs. (S3) and (S4). Application of the global humidity cor-   35
rection to the ERA5 HRES leads to (i) a smaller ISSR coverage area around the tropics, (ii) larger ISSR coverage above 40° N and below 40° S, and (iii) a higher occurrence of localised regions with RHi above 140 % (Fig. S5 in the Supplement). Further details on the extended humidity correction   40
are listed in Sect. S1.3 in the Supplement.

## 2.3 Contrail simulation

CoCiP simulates the properties and lifecycle of individual contrail segments along a flight trajectory (Schumann, 2012; Schumann et al., 2012). Two consecutive flight waypoints that satisfy the SAC form a contrail segment (Schumann, 1996), and persistent contrail segments are defined when their post-wake vortex ice water content (IWC) is greater than $10^{-12}\,\mathrm{kg\,kg^{-1}}$. The initial ice crystal number per contrail length ($n_{\mathrm{ice,\,initial}}$) is calculated by

$$n_{\mathrm{ice,\,initial}} = \max\left(\mathrm{nvPM\,EI_n},\, 10^{13}\right) \times \dot{m}_{\mathrm{f,dist}}$$
$$\times\, p_{\mathrm{activation}} \times f_{\mathrm{surv}}, \tag{5}$$

where

$$p_{\mathrm{activation}} = -0.661\exp(\mathrm{d}T_{\mathrm{SAC}}) + 1, \tag{6}$$
$$\mathrm{d}T_{\mathrm{SAC}} = T_{\mathrm{w}} - T_{\mathrm{SAC}} \;(< 0,\ \mathrm{in\ K}), \tag{7}$$

and

$$f_{\mathrm{surv}} = \frac{\mathrm{IWC_1}}{\mathrm{IWC_0}}. \tag{8}$$

The lower bound of nvPM $\mathrm{EI_n}$ is constrained to $10^{13}\,\mathrm{kg^{-1}}$ to account for the potential activation of semi-volatile particles and ambient aerosols (Kärcher, 2018). $\dot{m}_{\mathrm{f,dist}}$ is the fuel consumption per distance flown, $p_{\mathrm{activation}}$ is the proportion of nvPM that activates to form contrail ice crystals (Bräuer et al., 2021a; Teoh et al., 2022), and $f_{\mathrm{surv}}$ is the difference in IWC before and after the wake vortex phase as denoted by the subscripts "0" and "1", respectively. Contrail evolution is simulated using a Runge–Kutta scheme with model time steps of 300 s, which is a time resolution that is higher than previous CoCiP studies (1800–3600 s) that were constrained by computational demands (Schumann, 2012; Schumann et al., 2015; Teoh et al., 2020, 2022). Persistent contrail segments reach their end of life when the (i) contrail ice number concentration falls below background levels of $10^3\,\mathrm{m^{-3}}$; (ii) contrail optical depth ($\tau_{\mathrm{contrail}}$) is lower than $10^{-6}$; (iii) contrail age exceeds the maximum contrail lifetime that is set to 12 h; or (iv) midpoint of the contrail plume falls outside the defined altitude domain of between 6 and 13 km. The 13 km upper bound in condition (iv) was previously applied in Teoh et al. (2020) and could lead to a small underestimation of persistent contrail formation by military aircraft and private jets in the tropics, where 0.2 % of the global annual flight distance flown in GAIA occurred above 13 km and between $-30°$ S and 30° N. The local contrail RF (RF′) for each contrail segment, i.e. the change in radiative flux over the area covered by the contrail, is estimated using a parametric RF model, wherein the simulated contrail properties from CoCiP and meteorology from the ERA5 HRES are served as inputs.

CoCiP is set up in its original form without accounting for humidity exchange between contrails and the atmosphere and without radiative effects from contrail–contrail overlapping (Schumann, 2012; Teoh et al., 2020, 2022). Previous studies estimated that the annual mean contrail net RF could reduce by (i) 15 %–20 % when CoCiP was coupled with a general circulation model to account for the contrail–atmosphere humidity exchange and (ii) 3 % globally (or up to 57 % in regions with dense air traffic) when the effects from contrail–contrail overlapping are included (Schumann et al., 2015, 2021; Sanz-Morère et al., 2021). CoCiP has also been updated to incorporate the radiative heating effects on the contrail plume, where ice crystal losses are enhanced by the cumulative radiative energy absorbed by the contrail, which increases plume temperature and suppresses ice supersaturation, and by the differential heating rate, which drives convective turbulence and vertical mixing (Jensen et al., 1998; Schumann and Graf, 2013; Schumann and Heymsfield, 2017; Schumann et al., 2010). Both quantities are estimated in accordance with Schumann et al. (2010), who developed a parametric model fitted to outputs from the libRadtran radiative transfer model (Mayer and Kylling, 2005).

We note that the CoCiP algorithm has recently been released as open-source code and can be accessed via the pycontrails repository on GitHub (Shapiro et al., 2023), with the global contrail simulations in this study conducted using pycontrails v0.30.0. The regional contrail properties and climate forcing are estimated using rectangular spatial bounding boxes (Fig. 2 and Table S5) that are consistent with previous studies (Wilkerson et al., 2010; Hoare, 2014; Teoh et al., 2024). The five different output formats provided by CoCiP are described in Sect. S2.

## 2.4 Climate forcing metrics

Five different metrics are used to report the contrail climate forcing. Firstly, the contrail SW and LW RF′ is calculated from a parametric RF model using inputs of the simulated contrail properties and ambient meteorology (Schumann et al., 2012). Secondly, the contrail energy forcing ($\mathrm{EF_{contrail}}$) provides the cumulative contrail climate forcing over its lifetime (Schumann et al., 2011),

$$\mathrm{EF_{contrail}\,[J]} = \int_0^T \mathrm{RF'_{net}}(t) \times L(t) \times W(t)\,\mathrm{d}t, \tag{9}$$

where $T$, $L$, and $W$ are the contrail segment lifetime, length, and width, respectively. Thirdly, the annual mean contrail RF quantifies the change in radiative flux over the globe/region at a given time, and it is estimated from the annual $\mathrm{EF_{contrail}}$, assuming a linear relationship between contrail cover and RF,

$$\text{Annual mean contrail net RF}\,[\mathrm{W\,m^{-2}}] =$$
$$\frac{\sum \mathrm{EF_{contrail}\,[J]}}{S_{\mathrm{region}}\,[\mathrm{m^2}] \times \left(365 \times 24 \times 60^2\,[\mathrm{s}]\right)}, \tag{10}$$

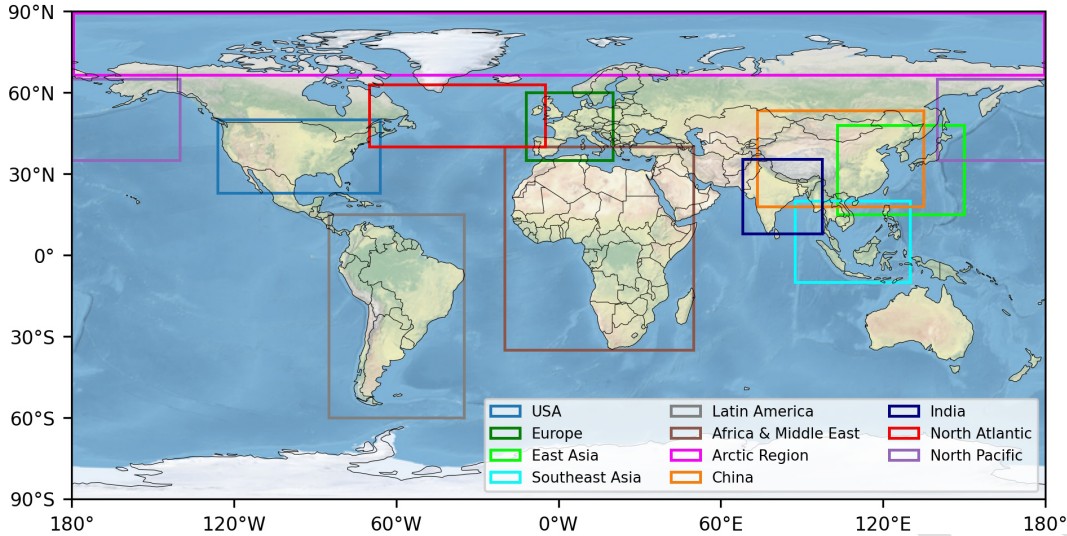

**Figure 2.** Spatial bounding boxes used to estimate the regional air traffic, emissions, and contrail properties. The specific dimensions of these bounding boxes can be found in Table S5 in the Supplement. Basemap plotted using Cartopy 0.22.0 and sourced from Natural Earth; licensed under public domain.

where $S_{\mathrm{region}}$ is the surface area of the region of interest. As CoCiP simulates the full lifecycle of contrails (from line-shaped clouds to contrail cirrus) and does not simulate any second-order effects, our climate forcing estimates are comparable to estimates of RF due to "contrail cirrus" in previous studies.

Next, the global annual mean contrail ERF is estimated from the RF by assuming a mean ERF / RF ratio of 0.42 (Lee et al., 2021). The ERF accounts for the rapid atmospheric adjustments (i.e. atmosphere–humidity exchange and temperature lapse rate) and natural cirrus responses (i.e. reduction in natural cirrus occurrence and cloudiness) resulting from the contrail (Lee et al., 2023). Thus, the ERF / RF ratio is a measure of efficacy which describes how effective the contrail RF impacts the global mean surface temperature compared to the $CO_2$-induced RF (Myhre et al., 2013). Our assumed ERF / RF ratio (0.42) is based on three global climate model studies that estimate it to range between 0.31 and 0.59 (Ponater et al., 2005; Rap et al., 2010; Bickel et al., 2019), although a lower ERF / RF ratio of 0.21 was estimated from a recent coupled atmosphere–ocean climate model (Bickel, 2023). Due to the large uncertainty and spatiotemporal variabilities in the contrail efficacy (Ponater et al., 2005; Schumann and Mayer, 2017; Gettelman et al., 2021), we (i) base our analysis on the instantaneous contrail climate forcing (RF and $EF_{\mathrm{contrail}}$) and (ii) only apply the ERF / RF conversion at a global scale rather than on individual flights, focusing solely on comparing our global annual mean contrail ERF with existing studies (Lee et al., 2021).

Finally, we also approximate the contrail cirrus global warming potential over a 20-year ($GWP_{20}$) and 100-year ($GWP_{100}$) time horizon using the energy forcing metric,

$$\text{Contrail cirrus GWP}_{\mathrm{TH}} = \frac{EF_{\mathrm{contrail}} \times \left(\frac{ERF}{RF}\right)}{EF_{CO_2,\mathrm{TH}}}. \quad (11)$$

The ERF / RF ratio is assumed to be 0.42, and the $CO_2$ energy forcing ($EF_{CO_2,\mathrm{TH}}$) is calculated according to Teoh et al. (2020),

$$EF_{CO_2,\mathrm{TH}}\,[\mathrm{J}] = \int_0^{\mathrm{TH}} RF_{CO_2}\mathrm{d}t \times S_{\mathrm{Earth}}$$

$$\approx \text{AGWP}_{CO_2,\mathrm{TH}} \times \left(365 \times 24 \times 60^2\right)$$

$$\times m_{CO_2} \times S_{\mathrm{Earth}}, \quad (12)$$

where $S_{\mathrm{Earth}}$ is the Earth surface area ($5.101 \times 10^{14}\,\mathrm{m}^2$), $m_{CO_2}$ is the total $CO_2$ emissions, and the $CO_2$ absolute global warming potential over 20-years ($\text{AGWP}_{CO_2,20}$) and 100-years ($\text{AGWP}_{CO_2,100}$) are assumed to be $2.39 \times 10^{-14}$ and $88.0 \times 10^{-15}\,\mathrm{yr\,W\,m^{-2}\,kg^{-1}}$, respectively (Gaillot et al., 2023).

## 2.5 Sensitivity analysis

Earlier studies suggest that the simulated contrail properties are highly sensitive to the humidity fields, aircraft performance (fuel consumption and overall efficiency), nvPM particle number emissions, and contrail model parameters (Schumann et al., 2021; Teoh et al., 2022; Bier and Burkhardt, 2022). However, these sensitivity studies are limited, because the simulations only cover a specific region (Schumann et al., 2021; Teoh et al., 2022), were re-run using

different constants of soot $EI_n$ (Bier and Burkhardt, 2022), or assume the default aircraft–engine assignment that is provided by BADA (Schumann et al., 2021; Teoh et al., 2022, 2020).

To assess the sensitivity of CoCiP to various inputs and contrail model parameters, we perform a sensitivity analysis by re-running the global contrail simulation for 2019 with seven distinct set-ups: (i) a simulation without humidity corrections applied to the ERA5 HRES (Sect. 2.2); (ii) a simulation using a constant humidity correction that was adopted in earlier studies (Schumann, 2012; Schumann et al., 2015; Teoh et al., 2020; Schumann et al., 2021), where the ERA5-derived RHi fields were uniformly increased by dividing it with a factor of 0.95; (iii) a simulation that uses the default aircraft–engine combination from BADA3 (EURO-CONTROL, 2019) instead of the specific aircraft variant and engine model provided by a fleet database (Cirium, 2022); two simulations where all waypoints are assumed with a constant nvPM $EI_n$ of (iv) $10^{15}\,\mathrm{kg^{-1}}$ and (v) $10^{14}\,\mathrm{kg^{-1}}$, respectively; (vi) a simulation without the effects of radiative heating interactions with the contrail plume; and (vii) a simulation that approximates the change in contrail climate forcing due to contrail–contrail overlapping (methodology detailed in Sect. S4.3).

Sensitivity experiments (i), (iii), (vi), and (vii) are set up to assess the impact of improved input parameters and updates to the contrail modelling processes on the simulated contrail climate forcing (Schumann et al., 2021, 2010), while sensitivity experiments (ii), (iii), and (iv) are designed to align with the methodology of previous studies and explore their potential implications (Schumann, 2012; Schumann et al., 2015; Teoh et al., 2020; Bier and Burkhardt, 2022). In sensitivity experiment (v), the nvPM $EI_n$ is fixed at the threshold marking the transition from soot-rich to soot-poor conditions ($\sim 10^{14}\,\mathrm{kg^{-1}}$) to estimate the minimum contrail climate forcing that could be achieved through reductions in aircraft nvPM emissions.

## 3 Results and discussion

Section 3.1 presents the simulated global and regional contrail properties, including the annual statistics for 2019 (Sect. 3.1.1) and 2020–2021 (Sect. 3.1.2), seasonal effects (Sect. 3.1.3), and the spatiotemporal patterns and set of conditions that cause strongly warming/cooling contrail segments (Sect. 3.1.4). Section 3.2 evaluates the sensitivity of the 2019 contrail climate forcing to different input and model parameters, while Sect. 3.3 compares our results with existing studies. Additional data, tables, and statistics from the global contrail simulation that are not presented here can be found in the Supplement as referenced in the text.

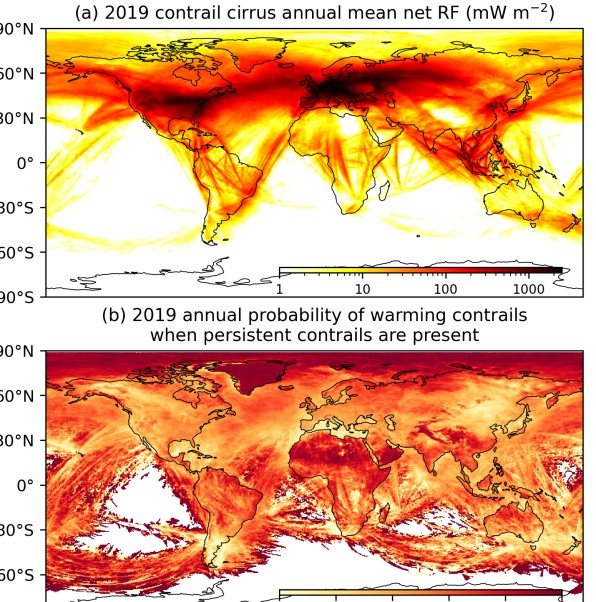

**Figure 3.** The 2019 global **(a)** annual mean contrail cirrus net RF and **(b)** annual probability of warming contrails, where the hourly contrail net RF is greater than zero when persistent contrails are present in the grid cell. The global annual mean contrail cirrus SW and LW RF and the annual mean ratio of contrail LW-to-SW RF are shown in Fig. S7 in Sect. S3. Basemap plotted using Cartopy 0.22.0 and sourced from Natural Earth; licensed under public domain.

### 3.1 Global contrail simulation

#### 3.1.1 2019 global and regional contrail climate forcing

In 2019, 40.2 million flights collectively flew $60.9 \times 10^9\,\mathrm{km}$, of which 24 % of flights and 5 % of the annual distance flown formed persistent contrails (Table 1). The fleet-aggregated mean nvPM $EI_n$ and nvPM per flight distance flown are $1.02 \times 10^{15}\,\mathrm{kg^{-1}}$ and $4.7 \times 10^{12}\,\mathrm{m^{-1}}$, respectively, and around 53 % of the nvPM activated to form contrail ice crystals and persisted after the wake vortex phase ($2.5 \times 10^{12}\,\mathrm{m^{-1}}$) (Table 1). We estimate that these persistent contrail segments have the following mean values: (i) lifetime of 2.4 h; (ii) dimension of 9900 m (width) $\times$ 800 m (depth); (iii) ice particle volume mean radius ($r_{ice}$) of $\sim 10\,\mathrm{\mu m}$; and (iv) $\tau_{contrail}$ of 0.14, respectively. Globally, contrail cirrus covers 0.06 % of the sky area and the annual mean LW RF ($126\,\mathrm{mW\,m^{-2}}$) is around 2 times larger than the SW RF ($-63.7\,\mathrm{mW\,m^{-2}}$), yielding a net RF of $62.1\,\mathrm{mW\,m^{-2}}$ (Table 1 and Fig. 3a).

Regional variabilities in the annual mean contrail cirrus net RF can be explained by differences in the: (i) annual flight distance flown, (ii) percentage of flight distance forming persistent contrails ($p_{contrail}$), and (iii) $EF_{contrail}$ per unit length of contrail (Fig. 4). The USA and Europe have the largest contrail climate forcing, accounting for 21 % and 18 % of the

**Table 1.** Global annual air traffic activity, emissions, and contrail properties from 2019 to 2021. TS2

| Annual statistics | Units | 2019 | 2020 | 2021 | 2019 vs. 2020 | 2019 vs. 2021 |
|---|---|---|---|---|---|---|
| Number of flights | – | 40 220 293 | 27 910 301 | 35 576 165 | −31 % | −12 % |
| Annual flight distance flown | $10^9$ km | 60.94 | 34.50 | 41.90 | −43 % | −31 % |
| Annual fuel burn | $10^9$ kg | 280 | 145 | 165 | −48 % | −41 % |
| Fuel burn per flight distance | kg km$^{-1}$ | 4.596 | 4.200 | 3.926 | −8.6 % | −15 % |
| Annual $CO_2$ emissions | $10^9$ kg | 885 | 458 | 520 | −48 % | −41 % |
| Mean overall efficiency, $\eta$ | – | 0.297 | 0.285 | 0.289 | −4.0 % | −2.7 % |
| Mean nvPM $EI_n$ | $10^{15}$ kg$^{-1}$ | 1.021 | 1.016 | 1.021 | −0.5 % | 0.0 % |
| Mean nvPM per flight distance | $10^{12}$ m$^{-1}$ | 4.693 | 4.265 | 4.009 | −9.1 % | −15 % |
| Flights forming contrails | % | 42.5 | 34.6 | 34.2 | −19 % | −20 % |
| Flights forming persistent contrails[a] | % | 23.8 | 17.7 | 17.8 | −25 % | −25 % |
| Annual contrail length | $10^9$ km | 21.35 | 11.29 | 14.00 | −47 % | −34 % |
| Flight distance forming contrails | % | 35.0 | 32.7 | 33.4 | −6.6 % | −4.6 % |
| Annual persistent contrail length[a] | $10^9$ km | 3.018 | 1.403 | 1.732 | −54 % | −43 % |
| Flight distance forming persistent contrails[a] | % | 4.95 | 4.07 | 4.13 | −18 % | −17 % |
| Initial mean ice particle number per contrail length, $n_{ice,\,initial}$ | $10^{12}$ m$^{-1}$ | 2.50 | 2.31 | 2.17 | −8 % | −13 % |
| Lifetime mean ice particle number per contrail length, $n_{ice}$ | $10^{12}$ m$^{-1}$ | 1.88 | 1.75 | 1.64 | −6.9 % | −13 % |
| Mean contrail lifetime | h | 2.43 | 2.34 | 2.25 | −3.6 % | −7.3 % |
| Mean ice particle volume mean radius, $r_{ice}$ | μm | 9.96 | 10.4 | 10.7 | 4.4 % | 7.4 % |
| Mean contrail segment optical depth, $\tau_{contrail}$ | – | 0.139 | 0.127 | 0.126 | −8.4 % | −9.2 % |
| Mean contrail width | m | 9903 | 9573 | 9081 | −3.3 % | −8.3 % |
| Mean contrail depth | m | 803 | 782 | 776 | −2.6 % | −3.4 % |
| Contrail cirrus coverage[b] | % | 0.064 | 0.030 | 0.038 | −53 % | −41 % |
| Contrail cirrus coverage, clear sky[c] | % | 0.66 | 0.28 | 0.33 | −57 % | −50 % |
| Cloud–contrail overlap | % | 90.3 | 89.3 | 88.5 | −1.1 % | −2.0 % |
| Number of flights: warming contrails | – | 6 741 548 | 3 491 439 | 4 430 717 | −48 % | −34 % |
| Number of flights: cooling contrails | – | 2 821 562 | 1 458 930 | 1 904 533 | −48 % | −33 % |
| Ratio: warming-to-cooling contrails | – | 2.39 | 2.39 | 2.33 | 0.2 % | −2.6 % |
| Mean contrail cirrus SW RF′ | W m$^{-2}$ | −4.15 | −3.70 | −3.90 | −11 % | −6.0 % |
| Mean contrail cirrus LW RF′ | W m$^{-2}$ | 5.36 | 4.89 | 5.05 | −8.8 % | −5.8 % |
| Mean contrail cirrus net RF′ | W m$^{-2}$ | 1.22 | 1.18 | 1.15 | −3.3 % | −5.7 % |
| Annual mean contrail cirrus SW RF | mW m$^{-2}$ | −63.7 | −26.4 | −33.0 | −59 % | −48 % |
| Annual mean contrail cirrus LW RF | mW m$^{-2}$ | 126 | 53.8 | 64.8 | −57 % | −49 % |
| Annual mean contrail cirrus net RF | mW m$^{-2}$ | 62.1 | 27.3 | 31.7 | −56 % | −49 % |
| Annual mean contrail cirrus net ERF | mW m$^{-2}$ | 26.1 | 11.5 | 13.3 | −56 % | −49 % |
| Annual $EF_{contrail}$ | $10^{18}$ J | 999 | 440 | 510 | −56 % | −49 % |
| $EF_{contrail}$ per flight distance | $10^8$ J m$^{-1}$ | 0.164 | 0.128 | 0.122 | −22 % | −26 % |
| $EF_{contrail}$ per contrail length | $10^8$ J m$^{-1}$ | 3.31 | 3.14 | 2.94 | −5 % | −11 % |
| Flights responsible for 80 % $EF_{contrail}$ | % | 2.68 | 1.78 | 1.73 | −34 % | −35 % |
| Contrail cirrus $GWP_{20}$[d] | – | 1.17 | 0.99 | 1.02 | −15 % | −13 % |
| Contrail cirrus $GWP_{100}$[d] | – | 0.32 | 0.27 | 0.28 | −15 % | −13 % |

[a] Persistent contrails are defined when the contrail ice water content after the wake vortex phase is greater than $10^{-12}$ kg kg$^{-1}$. We note that contrails do not sublimate immediately when the RHi is less than 100 %, as the ice crystal sublimation rate depends on the background RHi. [b] Global contrail cirrus cover as a percentage of sky area. Contrail cirrus is assumed to be present in a grid cell if (i) the weighted-sum of the contrail optical depth is greater than 0.1, which is selected to be consistent with the satellite detectability threshold, and (ii) the summation of the natural cirrus optical depth in the grid cell is less than 0.1. [c] Global contrail cirrus coverage area under clear-sky conditions without the presence of natural cirrus. Contrails are present in a grid cell if the weighted-sum of the contrail optical depth is greater than 0.1. [d] The contrail cirrus GWP over a 100-year and 20-year time horizon is approximated using the energy forcing metric as described in Eqs. (11) and (12).

**Table 2.** Regional air traffic activity, emissions, and contrail properties for 2019. The statistics for 2020 and 2021 can be found in Tables S6 and S7 (Sect. S3).

| Regional statistics: 2019 | Global | USA | Europe | East Asia | SEA | Latin America | Africa & Middle East | China | India | North Atlantic | North Pacific | Arctic Region |
|---|---|---|---|---|---|---|---|---|---|---|---|---|
| Annual distance flown ($\times 10^9$ km) | 60.94 | 16.30 | 8.858 | 8.303 | 3.989 | 2.250 | 4.631 | 8.946 | 2.551 | 2.975 | 2.393 | 0.3826 |
| Percentage relative to global values[a] | – | 26.8 % | 14.5 % | 13.6 % | 6.5 % | 3.7 % | 7.6 % | 14.7 % | 4.2 % | 4.9 % | 3.9 % | 0.6 % |
| Annual dist. flown above FL250 ($\times 10^9$ km) | 49.03 | 11.97 | 6.779 | 6.477 | 3.207 | 1.803 | 3.952 | 7.091 | 2.134 | 2.876 | 2.033 | 0.3738 |
| Percentage relative to global values[a] | – | 24.4 % | 13.8 % | 13.2 % | 6.5 % | 3.7 % | 8.1 % | 14.5 % | 4.4 % | 5.9 % | 4.1 % | 0.8 % |
| Air traffic density (km$^{-1}$ h$^{-1}$) | 0.014 | 0.116 | 0.152 | 0.059 | 0.029 | 0.006 | 0.009 | 0.047 | 0.032 | 0.030 | 0.012 | 0.002 |
| Mean nvPM EI$_n$ ($\times 10^{15}$ kg$^{-1}$) | 1.021 | 1.361 | 1.155 | 1.082 | 0.983 | 1.035 | 0.883 | 1.083 | 1.036 | 0.621 | 0.649 | 0.406 |
| Mean nvPM per dist. ($\times 10^{12}$ m$^{-1}$) | 4.69 | 4.48 | 4.79 | 5.44 | 4.83 | 4.37 | 4.44 | 5.41 | 5.36 | 3.93 | 4.53 | 3.18 |
| Persistent contrail length ($\times 10^9$ km) | 3.018 | 0.807 | 0.646 | 0.137 | 0.133 | 0.073 | 0.128 | 0.175 | 0.045 | 0.308 | 0.136 | 0.025 |
| Percentage relative to global values[a] | – | 26.7 % | 21.4 % | 4.5 % | 4.4 % | 2.4 % | 4.2 % | 5.8 % | 1.5 % | 10.2 % | 4.5 % | 0.8 % |
| Dist. forming persistent contrails | 4.95 % | 4.95 % | 7.29 % | 1.65 % | 3.34 % | 3.25 % | 2.75 % | 1.96 % | 1.75 % | 10.4 % | 5.68 % | 6.64 % |
| Area-mean contrail optical depth, $\tau$ | 0.018 | 0.065 | 0.099 | 0.027 | 0.020 | 0.014 | 0.013 | 0.026 | 0.019 | 0.051 | 0.023 | 0.026 |
| Mean contrail age in domain (h) | 2.43 | 1.84 | 1.87 | 2.31 | 2.72 | 3.44 | 2.81 | 2.23 | 2.25 | 2.26 | 2.60 | 3.51 |
| Contrail cirrus coverage (%) | 0.06 | 0.31 | 0.95 | 0.05 | 0.03 | 0.01 | 0.02 | 0.99 | 0.02 | 0.36 | 0.07 | 0.05 |
| Contrail cirrus coverage, clear sky (%) | 0.66 | 5.4 | 8.7 | 1.0 | 0.88 | 0.082 | 0.14 | 0.90 | 0.38 | 4.0 | 0.54 | 0.12 |
| Annual mean SW RF (mW m$^{-2}$) | −63.7 | −485 | −1160 | −88.9 | −83.8 | −14.7 | −20.0 | −87.8 | −35.6 | −300 | −55.0 | −10.2 |
| Annual mean LW RF (mW m$^{-2}$) | 126 | 900 | 2038 | 153 | 174 | 33.3 | 38.7 | 150 | 81.2 | 601 | 103 | 29.2 |
| Annual mean net RF (mW m$^{-2}$) | 62.1 | 414 | 876 | 63.9 | 90.4 | 18.5 | 18.6 | 62.3 | 45.4 | 300 | 47.7 | 19.0 |
| Ratio: LW / SW RF | 1.98 | 1.86 | 1.76 | 1.72 | 2.08 | 2.27 | 1.94 | 1.71 | 2.28 | 2.00 | 1.87 | 2.86 |
| EF$_{contrail}$ ($\times 10^{18}$ J) | 999 | 209 | 184 | 32.6 | 44.2 | 23.3 | 35.4 | 42.5 | 13.2 | 109 | 35.7 | 12.9 |
| Percentage relative to global values[a] | – | 20.9 % | 18.4 % | 3.3 % | 4.4 % | 2.3 % | 3.5 % | 4.3 % | 1.3 % | 10.9 % | 3.6 % | 1.3 % |
| EF$_{contrail, initial}$ ($\times 10^{18}$ J)[b] | 999 | 218 | 201 | 32.9 | 44.8 | 23.6 | 37.8 | 41.4 | 13.2 | 116 | 35.0 | 10.2 |
| Percentage relative to global values[a] | – | 21.8 % | 20.1 % | 3.3 % | 4.5 % | 2.4 % | 3.8 % | 4.1 % | 1.3 % | 11.6 % | 3.5 % | 1.0 % |
| Ratio: EF$_{contrail}$ / EF$_{contrail, initial}$[c] | 1.00 | 0.96 | 0.92 | 0.99 | 0.99 | 0.99 | 0.94 | 1.03 | 1.00 | 0.94 | 1.02 | 1.26 |
| EF$_{contrail}$ per flight distance ($\times 10^8$ J m$^{-1}$) | 0.164 | 0.134 | 0.227 | 0.040 | 0.112 | 0.105 | 0.082 | 0.046 | 0.052 | 0.390 | 0.146 | 0.267 |
| EF$_{contrail}$ per contrail length ($\times 10^8$ J m$^{-1}$) | 3.31 | 2.70 | 3.11 | 2.40 | 3.36 | 3.23 | 2.96 | 2.37 | 2.95 | 3.76 | 2.57 | 4.02 |

[a] There is some overlapping between regional bounding boxes (Fig. 2); therefore, the summation of regional statistics does not add up to 100 %. [b] The total EF$_{contrail}$ throughout the contrail lifetime is added back to the location where contrails were initially formed. [c] A higher ratio indicates that a larger share of contrail climate forcing is from contrails initially formed outside of the region but subsequently advected into the domain.

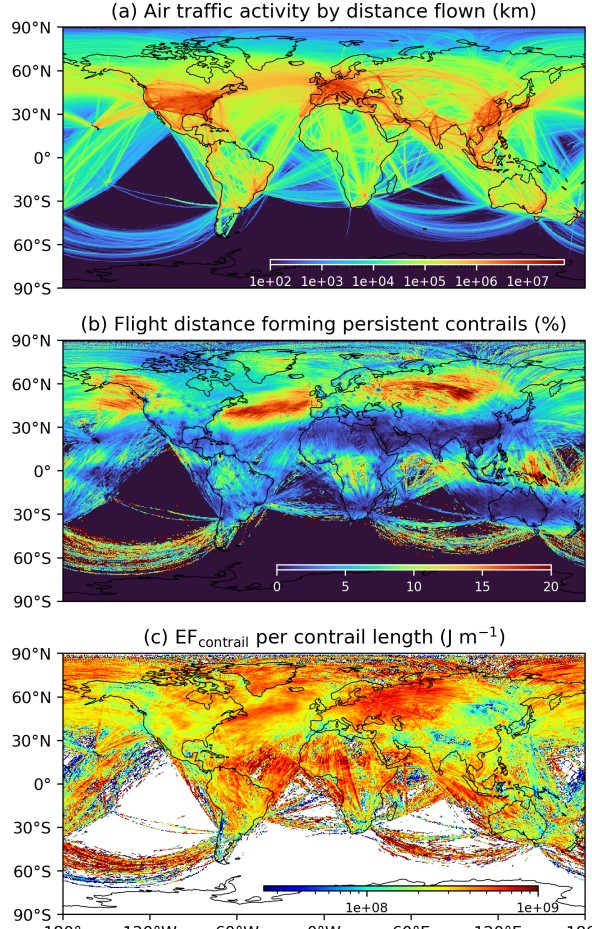

**Figure 4.** The 2019 global **(a)** annual flight distance flown, **(b)** percentage of flight distance forming persistent contrails, and **(c)** $EF_{contrail}$ per unit length of persistent contrail formed, where the total $EF_{contrail}$ throughout the contrail lifetime is added to the pixel where contrails were initially formed. Basemap plotted using Cartopy 0.22.0 and sourced from Natural Earth; licensed under public domain.

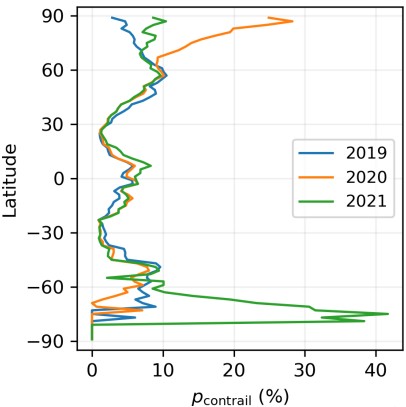

**Figure 5.** The percentage of annual flight distance flown that formed persistent contrails ($p_{contrail}$) by latitude in 2019 (blue line), 2020 (orange line), and 2021 (green line). Several factors collectively contribute to the large inter-annual variability in $p_{contrail}$ at high latitudes (above 60° N and below 60° S), including the (i) smaller grid cell area at high latitudes, which can cause a larger inter-annual variability in the ISSR occurrence relative to other latitude bands (see Fig. S14), and (ii) low air traffic activity at high latitudes where 0.62 % and 0.06 % of the global annual flight distance were flown at latitudes above 66.5° N and below 45° S, respectively (Teoh et al., 2024), thereby causing $p_{contrail}$ at these latitude bins to be calculated from a significantly smaller sample size relative to other latitudes.

global annual $EF_{contrail}$, respectively, because they have the highest air traffic activity (27 % and 15 % of the global annual flight distance flown, respectively) (Table 2). We note that the annual mean contrail net RF over Europe (876 mW m$^{-2}$) is around 2 times larger than the USA (414 mW m$^{-2}$) because (i) Europe is situated at a higher latitude which likely caused $p_{contrail}$ (7.3 %) to be larger than the USA (5.0 %) (Table 2) and (ii) the contrail forcing in Europe is concentrated over a smaller domain area (cf. Eq. 10 and Fig. 2).

The North Atlantic has a significantly higher share of annual $EF_{contrail}$ (11 %) relative to its flight distance flown (4.9 %); this is because flights are predominantly flown at cruising altitudes (Fig. S8) and because of the influence of warm conveyor belts in transporting humid air to cruise altitudes, which can lead to larger ISSR coverage area in this region (Voigt et al., 2017). Both factors likely caused $p_{contrail}$

($\sim 10$ %) in the North Atlantic to be 2 times higher than the global average ($\sim 5$ %). In contrast, the share of annual $EF_{contrail}$ over China and India (6 %) is significantly lower than their flight distance flown (19 %), and their contrail net RF values (62–64 mW m$^{-2}$) are close to the global net RF (62.1 mW m$^{-2}$) (Table 2). This phenomenon is likely caused by the Hadley Circulation (where warm and moist air around the surface of the Equator rises to the upper troposphere, moves poleward, becomes drier and cooler, and sinks at the subtropics) and lower cruising altitudes (Fig. S8), both of which likely reduced the flight distance flown in ISSRs (Lau and Kim, 2015; Reutter et al., 2020) and causes $p_{contrail}$ in the subtropics ($\sim 2$ %) to be lower than the global average ($\sim 5$ %) (Fig. 5).

The $EF_{contrail}$ per contrail length also tends to be large over the Atlantic and Indian Ocean, Sahara, central Europe, and Greenland (Fig. 4c), and these effects can be attributed to (i) high mean albedos (Fig. S12), which reduces the contrail SW RF; (ii) high surface temperatures and outgoing longwave radiation (OLR) (Fig. S11), which drives the contrail LW RF; and/or (iii) flight scheduling, where long-haul flights tend to fly at night and at higher altitudes (Sect. 3.1.4). On average, contrails persisting over Greenland and the Sahara have a net warming effect in $\sim 90$ % of the hourly time periods, while contrails over the Mediterranean Sea and southern Argentina/Chile are cooling for $\sim 43$ % of the time (Fig. 3b).

### 3.1.2 Impacts from COVID-19

COVID-19 caused significant reductions in the global annual mean contrail cirrus net RF in 2020 ($27.3\,\mathrm{mW\,m^{-2}}$) and 2021 ($31.7\,\mathrm{mW\,m^{-2}}$) when compared to 2019 ($62.1\,\mathrm{mW\,m^{-2}}$). The percentage reduction in global contrail net RF ($-56\,\%$ in 2020 and $-49\,\%$ in 2021 relative to 2019) is greater than the change in global annual flight distance flown ($-43\,\%$ and $-31\,\%$, respectively) due to an increased share of (i) general aviation activity below $30\,000\,\mathrm{ft}$ ($9144\,\mathrm{m}$) (Teoh et al., 2024), which likely lowered $p_{\mathrm{contrail}}$ from $5.0\,\%$ in 2019 to $4.1\,\%$ in 2020 and 2021, and (ii) short-haul flights ($<3\,\mathrm{h}$) from $83\,\%$ in 2019 to $88\,\%$ in 2020 and 2021 (Teoh et al., 2024), where increased usage of narrow-body aircraft lowers the fleet-aggregated mean fuel consumption and nvPM per flight distance flown ($-9\,\%$ in 2020 and $-15\,\%$ in 2021 relative to 2019) and $\mathrm{EF}_{\mathrm{contrail}}$ per contrail length ($-5\,\%$ and $-11\,\%$, respectively).

A regional comparison between 2019 and 2020 shows that the North Atlantic experienced the largest percentage reduction in annual flight distance flown ($-61\,\%$) and contrail net RF ($-65\,\%$), because it has a large share of long-haul international flights which were most impacted by COVID-19. In East Asia and China, the reduction in contrail net RF ($50\,\%$–$54\,\%$) is more than 2 times greater than their reduction in flight distance flown ($-24\,\%$). The higher relative reduction in contrail net RF is most likely due to (i) a higher share of domestic air traffic in parts of China (Fig. 6a) that led to an $8\,\%$ increase in flight distance flown at $25\,000$–$30\,000\,\mathrm{ft}$ ($7620$–$9144\,\mathrm{m}$)TS3 where persistent contrails are less likely to form, and (ii) the reduction in international overflights caused a $39\,\%$ reduction in flight distance flown above $30\,000\,\mathrm{ft}$ ($9144\,\mathrm{m}$) (Fig. S9c). In contrast, the $11\,\%$ reduction in contrail net RF over the Arctic is significantly smaller than the $58\,\%$ reduction in flight distance flown, because a higher share of contrails in this region were formed elsewhere and subsequently advected into the domain (Table 2 vs. Table S6).

The 2021 annual mean contrail net RF values in 10 of the 11CE3 regions are $4\,\%$–$52\,\%$ larger than their 2020 levels (Table S6 vs. Table S7). There is a $44\,\%$ year-on-year reduction in the contrail net RF over the Arctic which is most likely caused by the inter-annual variability in meteorology (Teoh et al., 2022), where the 2020 $p_{\mathrm{contrail}}$ ($12\,\%$) and $\mathrm{EF}_{\mathrm{contrail}}$ per flight distance ($0.6 \times 10^8\,\mathrm{J\,m^{-1}}$) were outliers that were around 2 times larger than those recorded in 2019 ($6.6\,\%$ and $0.3 \times 10^8\,\mathrm{J\,m^{-1}}$, respectively) and 2021 ($7.3\,\%$ and $0.3 \times 10^8\,\mathrm{J\,m^{-1}}$) (Fig. 5, Tables 2, S6, and S7). Notably, the 2021 annual mean contrail net RF over the Gulf of Mexico and Caribbean Sea ($70$–$100°\,\mathrm{W}$, $5$–$35°\,\mathrm{N}$) is $3.8\,\%$ larger than their 2019 levels (Fig. 6d), and this is most likely caused by the increase in air traffic over the southern-USA–Mexico (Fig. 6b) where contrails were formed and advected southwards over its lifetime.

### 3.1.3 Seasonal statistics

The seasonal statistics reported here are biased towards the Northern Hemisphere where $92\,\%$ of the global annual flight distance was flown (Teoh et al., 2024). In 2019, global air traffic activity peaked in the summer (JJA) and was $14\,\%$ below peak levels during the winter months (DJF) (Fig. 7a). The seasonal changes in contrail properties show the opposite effect where $p_{\mathrm{contrail}}$ in wintertime is around 1.5 times larger than the summer ($5.9\,\%$ vs. $3.8\,\%$, Fig. 7b), and persistent contrails formed during this time have a larger mean lifetime (2.8 vs. 2.4 h in the summer, Fig. 7e), global coverage area ($0.07\,\%$ vs. $0.05\,\%$ in 2019, Fig. 7f), net RF (86 vs. $40\,\mathrm{mW\,m^{-2}}$, Fig. 7g), and $\mathrm{EF}_{\mathrm{contrail}}$ per contrail length (10 vs. $7.4 \times 10^7\,\mathrm{J\,m^{-1}}$, Fig. 7h). The larger contrail occurrence, lifetime, and climate forcing in wintertime can be attributed to (i) larger seasonal ISSR coverage area in the northern mid-latitudes ($30$–$60°\,\mathrm{N}$) (Teoh et al., 2022), a region accounting for $64\,\%$ of global annual flight distance flown; (ii) smaller initial RHi ($107\,\%$ vs. $110\,\%$ in summertime, Fig. S15d) and $\mathrm{d}T_{\mathrm{SAC}}$ ($-7$ vs. $-4\,\mathrm{K}$, Fig. S15e) both of which lowers specific humidity and the amount of condensable water available (cf. Eq. S2), leading to smaller $r_{\mathrm{ice}}$ (11 vs. $13\,\mathrm{\mu m}$, Fig. 7d) and longer lifetimes (2.8 vs. 2.4 h); (iii) higher percentage of cloud–contrail overlapping ($91\,\%$ vs. $88\,\%$, Fig. S15f); and (iv) shorter daylight hours. Factors (iii) and (iv) are expected to lower the contrail SW RF (Teoh et al., 2022). In contrast, the shorter contrail lifetime during the summer is likely due to persistent contrails forming at warmer temperatures with a smaller $\mathrm{d}T_{\mathrm{SAC}}$, thereby reducing $p_{\mathrm{activation}}$ and $n_{\mathrm{ice,initial}}$ (cf. Eqs. 5 and 6 and Fig. 7c), which result in larger $r_{\mathrm{ice}}$ (Fig. 7d) and ice crystal sedimentation rate. The larger mean overlying natural cirrus optical depth above contrails ($\tau_{\mathrm{cirrus}}$) in summertime (0.39 vs. 0.25 in winter, Fig. S15h) also contributes to a smaller contrail climate forcing relative to the winter months (Schumann et al., 2012; Teoh et al., 2022).

Due to the higher relative contrail climate forcing in wintertime, around one-third of the days in 2019 and 2021 (110–121 d) accounted for half of the global annual $\mathrm{EF}_{\mathrm{contrail}}$, while two-thirds of the days (236–243 d) caused $80\,\%$ of the annual $\mathrm{EF}_{\mathrm{contrail}}$ (Fig. 8a). In 2020, the global contrail climate forcing was further concentrated on a smaller number of days, where $23\,\%$ (83 d) and $53\,\%$ (195 d) of the days accounted for $50\,\%$ and $80\,\%$ of the annual $\mathrm{EF}_{\mathrm{contrail}}$, respectively (Fig. 8a), because the percentage of annual flight distance flown in the first quarter of 2020 ($37\,\%$), where contrails are expected to be strongly warming, was higher than those recorded in 2019 ($23\,\%$) and 2021 ($19\,\%$) and because of the significant reduction in global air traffic activity during the spring and summer (Fig. 7a), where the $\mathrm{EF}_{\mathrm{contrail}}$ per contrail length is at a minimum (Fig. 7h).

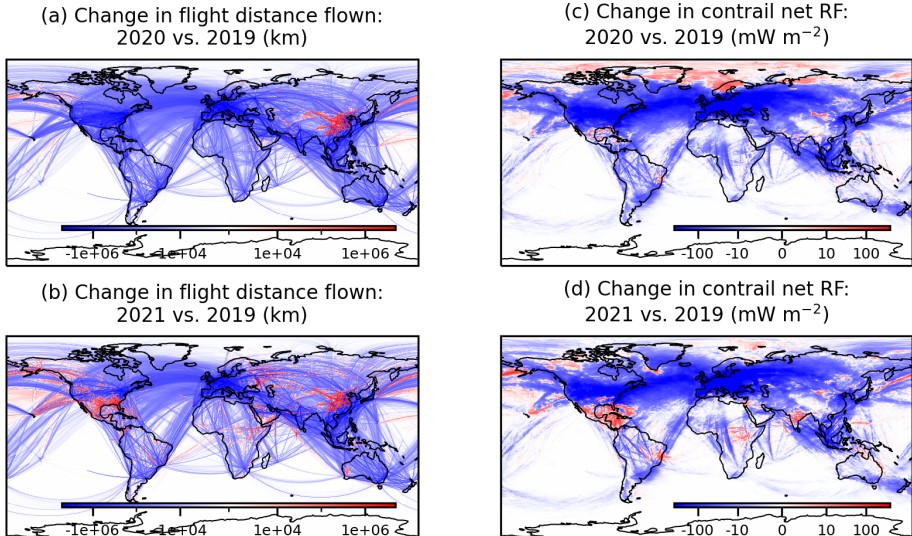

**Figure 6.** Change in the absolute annual flight distance flown when comparing between **(a)** 2020 vs. 2019 and **(b)** 2021 vs. 2019, and the annual mean contrail net RF between **(c)** 2020 vs. 2019 and **(d)** 2021 vs. 2019. Basemap plotted using Cartopy 0.22.0 and sourced from Natural Earth; licensed under public domain.

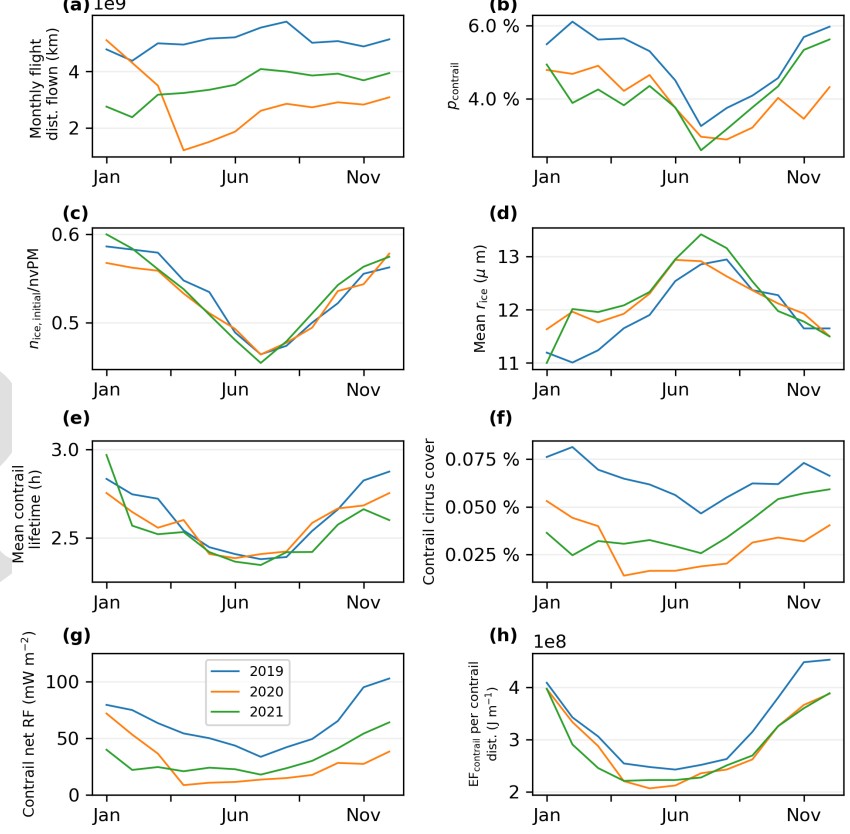

**Figure 7.** Monthly statistics on the global **(a)** flight distance flown, **(b)** percentage of flight distance forming persistent contrails ($p_{contrail}$), **(c)** fraction of nvPM that activates into contrail ice crystals and survive the wake vortex phase, **(d)** mean contrail ice particle volume mean radius ($r_{ice}$), **(e)** mean contrail lifetime, **(f)** global contrail cirrus coverage as a percentage of sky area, **(g)** mean net RF, and **(h)** $EF_{contrail}$ per unit length of persistent contrail formed from January 2019 to December 2021. Additional metrics that are not presented here are available in Fig. S15 in Sect. S3.

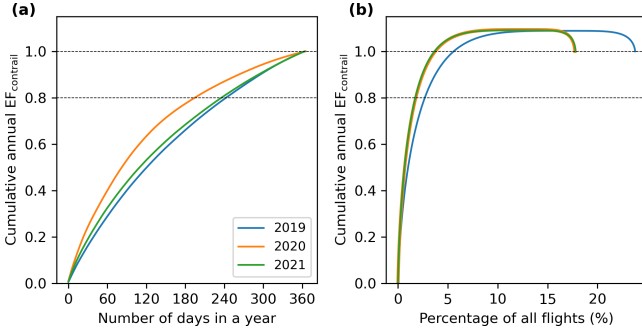

**Figure 8.** Cumulative density function of the global annual $EF_{contrail}$ from 2019 to 2021 versus the **(a)** number of days in a year and **(b)** the percentage of all flights that accounted for the proportion of $EF_{contrail}$. We note that the cumulative density function in **(b)** exceeds and returns to 1 because of the presence of flights with cooling contrails.

### 3.1.4 Strongly warming or cooling contrails

In 2019, 24 % of all flights formed persistent contrails, of which 70 % of these persistent contrail-forming flights have a net warming effect ($EF_{contrail} > 0$), and their mean magnitude of $EF_{contrail}$ ($1.45 \times 10^{14}$ J) is a factor of 5 larger than the remaining 30 % of contrail-forming flights with a net cooling effect ($-0.28 \times 10^{14}$ J). Around 2.7 % of all flights (or 11 % of contrail-forming flights) accounted for 80 % of the global annual $EF_{contrail}$ in 2019 (Fig. 8b). The contrail climate forcing was concentrated on a smaller subset of flights in 2020 and 2021, where 1.7 % of all flights or 10 % of contrail-forming flights accounted for 80 % of the annual $EF_{contrail}$ (Fig. 8b), and this is likely due to the larger share in general aviation activity and short-haul flights with flight times below 3 h (Teoh et al., 2024), both of which generally fly at lower altitudes where persistent contrails are less likely to form.

Individual flight segments with the most strongly warming contrails ($EF_{contrail}$ per contrail length $> 15.4 \times 10^8$ J m$^{-1}$, 95th percentile) are commonly found over the USA and North Atlantic (Fig. 9a), and these contrail segments are more likely to be formed by eastbound transatlantic flights and transcontinental flights across the USA (Table S9), because they tend to depart during the evenings (Teoh et al., 2022). In contrast, the most strongly cooling contrails ($EF_{contrail}$ per contrail length $< -2.39 \times 10^8$ J m$^{-1}$, 5th percentile) are more prevalent over Southeast Asia, North Asia, Europe, and the eastern North Atlantic region (Fig. 9b), and these contrail segments are more commonly formed by short- or medium-haul flights around Southeast Asia and East Asia; long-haul flights between Oceania, Asia, and Europe; and both eastbound and westbound transatlantic flights (Table S10). The ratio of solar direct radiation (SDR) to OLR also contributes to the spatial distribution of strongly warming and cooling contrail segments: strongly cooling contrails are more prevalent in Southeast Asia because the re-

gion has a high SDR-to-OLR ratio which drives the contrail SW RF and limits its LW RF; while the lower SDR-to-OLR ratio contributes to a higher share of strongly warming contrails at latitudes above 30° N (Fig. S13). Both strongly warming and cooling contrails are generally formed above 38 000 ft (11 580 m) over the tropics and across a wider altitude range (30 000–40 000 ft, 9144–12 192 m TS4) at higher latitudes (Figs. 9c and d), and this can likely be attributed to the latitude and seasonal variations in the tropopause height (Santer et al., 2003; Hoinka et al., 1993).

The most strongly warming contrail segments tend to occur during the winter and spring and between 16:00 and 03:00 LT (local time), while strongly cooling contrails are more prevalent in spring and between 03:00 and 12:00 LT (Fig. 9e and f). Both strongly warming and cooling contrails are formed by aircraft–engine types with mean nvPM emissions (6.9 and $5.3 \times 10^{12}$ m$^{-1}$, respectively) that are larger than the 2019 fleet-aggregated values ($4.7 \times 10^{12}$ m$^{-1}$, Table 1), and this is consistent with an earlier study that found a positive correlation between nvPM number emissions and the absolute magnitude and variability of $EF_{contrail}$ (Teoh et al., 2022). Strongly cooling contrails are more likely to have shorter lifetimes (mean of 5.6 h) relative to warming contrails (6.7 h) because of their smaller nvPM emissions and d$T_{SAC}$ (Fig. 9g and h), and their formation time (03:00–12:00 LT) and lifetime also suggest that these contrail segments spread during daylight hours and sublimate before dusk, thus maximising their SW RF.

The surface conditions and background cloud fields also influence the contrail climate forcing (Schumann et al., 2012; Teoh et al., 2022). Strongly warming contrail segments generally have a (i) larger effective albedo relative to cooling contrails (0.39 vs. 0.31); (ii) larger OLR (211 vs. 196 W m$^{-2}$); and (iii) smaller mean $\tau_{cirrus}$ (0.25 vs. 0.57) (Fig. 9i to k), and these are indicative of optically thick low-level water clouds, snowy and sandy terrains, and/or warmer surface temperatures. In contrast, contrails are more likely to be strongly cooling when formed over the dark ocean surface and/or below optically-thick high-level cirrus clouds that tend to reduce the contrail LW RF′ more strongly than the SW RF′ (Teoh et al., 2022).

### 3.2 Sensitivity analysis

We re-run the 2019 global contrail simulation to assess the sensitivity of the simulated contrail climate forcing to humidity corrections applied to the ERA5 HRES (Sect. 3.2.1), assumptions in aircraft–engine assignments and emissions (Sect. 3.2.2), and contrail model parameters (Sect. 3.2.3). Figure 10 and Table S11 summarises the change in global aviation emissions and contrail properties from the different simulation runs. When taken together, the sensitivity analysis estimates a 2019 global annual mean contrail cirrus net RF that falls within the range of 34.8 and 74.8 mW m$^{-2}$ (Table 3).

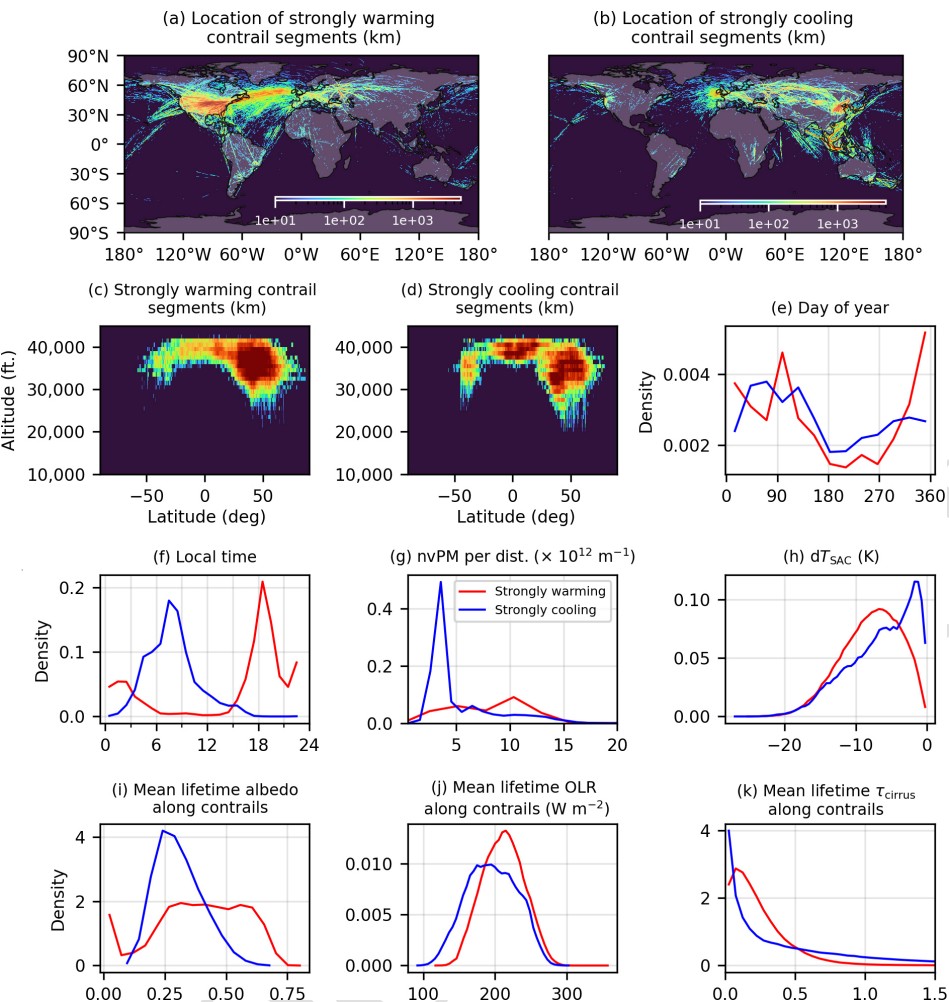

**Figure 9.** The location (longitude, latitude, and altitude) of individual contrail segments that are strongly warming ($EF_{contrail}$ per contrail length $> 15.4 \times 10^8$ J m$^{-1}$, 95th percentile) and strongly cooling ($EF_{contrail}$ per contrail length $< -2.39 \times 10^8$ J m$^{-1}$, 5th percentile), shown in panels **(a)** to **(d)**; and probability density functions showing their respective **(e)** day of year; **(f)** time of day; **(g)** nvPM emissions; **(h)** the $dT_{SAC}$ when these contrails were initially formed; and their mean **(i)** effective albedo; and **(j)** outgoing longwave radiation (OLR); and **(k)** $\tau_{cirrus}$ throughout their lifetime. Basemap plotted using Cartopy 0.22.0 and sourced from Natural Earth; licensed under public domain.

### 3.2.1  Humidity corrections

Globally, the baseline simulation with the extended global humidity correction (cf. Eqs. 1 to 4), produces an annual mean contrail net RF (62.1 mW m$^{-2}$) that is 78 % larger than the simulation without humidity correction applied to the ERA5 HRES (34.8 mW m$^{-2}$). The change in contrail net RF between the two simulations is largest at latitudes above 35° N (+96 %, 76.4 vs. 38.9 mW m$^{-2}$ without humidity correction), followed by the tropics (+59 %, 52.3 vs. 32.9 mW m$^{-2}$), and is smallest in the subtropics at around 30° N/S $\pm$ 5° (+2.2 %, 84.2 vs. 82.4 mW m$^{-2}$) (Fig. 10a).

Alternatively, the use of a constant humidity correction that was adopted in earlier studies (Schumann, 2012; Schumann et al., 2015; Teoh et al., 2020; Schumann et al., 2021) (cf. Eq. S5, where $RHi_c = 0.95$) causes the global annual

mean contrail net RF to be 4 % larger than the baseline simulation (64.5 vs. 62.1 mW m$^{-2}$). However, the constant humidity correction approach does not capture the latitude-dependent errors in the ERA5-derived ISSR (Table S1), and therefore, it could overestimate the annual mean contrail net RF in the tropics and subtropics and underestimate it at high latitudes (Fig. 10b).

### 3.2.2  Aircraft–engine assignment and emissions

The simulation with default aircraft–engine assignments from BADA3 causes the global annual mean contrail net RF to be 18 % larger than the baseline simulation (73.1 vs. 62.1 mW m$^{-2}$). This is because BADA3 assumes that some widely used aircraft types (i.e. Airbus A320, A320neo, and Boeing 787 families) are powered by engines with nvPM $EI_n$

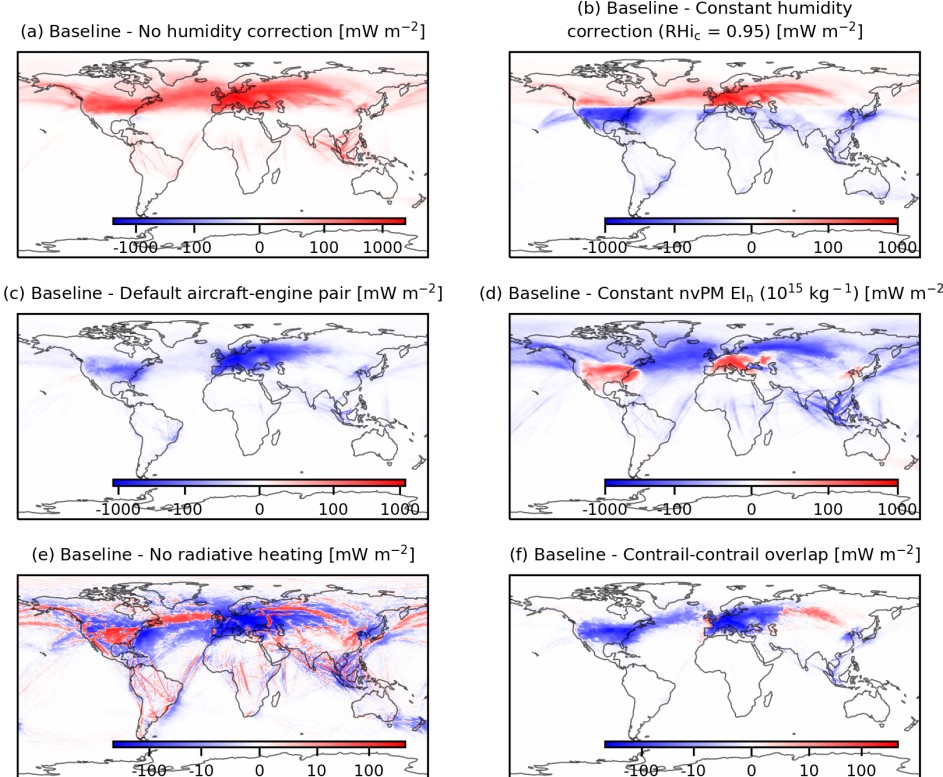

**Figure 10.** Absolute change in the 2019 global annual mean contrail cirrus net RF (in $\mathrm{mW\,m^{-2}}$) when comparing the baseline scenario with the simulation: **(a)** without humidity corrections applied to the ERA5 HRES; **(b)** with a constant humidity correction of RHi / $\mathrm{RHi_c}$, where $\mathrm{RHi_c} = 0.95$; **(c)** with the default aircraft–engine assignment from BADA3; **(d)** with a constant nvPM $\mathrm{EI_n}$ of $10^{15}\,\mathrm{kg^{-1}}$ for all waypoints; **(e)** without radiative heating effects; and **(f)** that approximates the radiative effects of contrail–contrail overlapping. Basemap plotted using Cartopy 0.22.0 and sourced from Natural Earth; licensed under public domain.

**Table 3.** Summary of the global contrail simulations performed in this study. Note that n/a represents not applicable CE4.

|  | Engine assignment | Humidity correction | Radiative heating effects | Contrail overlapping effects | Global annual mean contrail net RF ($\mathrm{mW\,m^{-2}}$) |
|---|---|---|---|---|---|
| Baseline simulations (2019–2021) | | | | | |
| 2019 | Cirium | Eqs. (1)–(4) | ✓ | × | 62.1 |
| 2020 | Cirium | Eqs. (1)–(4) | ✓ | × | 27.3 |
| 2021 | Cirium | Eqs. (1)–(4) | ✓ | × | 31.7 |
| Sensitivity: Humidity correction (2019) | | | | | |
| No humidity correction | Cirium | n/a | ✓ | × | 34.8 |
| Constant humidity correction | Cirium | Eq. (S5), $\mathrm{RHi_c} = 0.95$ | ✓ | × | 64.5 |
| Sensitivity: aircraft performance and emissions (2019) | | | | | |
| Default aircraft–engine assignment | BADA default | Eqs. (1)–(4) | ✓ | × | 73.1 |
| Constant nvPM $\mathrm{EI_n}$ ($10^{15}\,\mathrm{kg^{-1}}$) | n/a | Eqs. (1)–(4) | ✓ | × | 74.8 |
| Constant nvPM $\mathrm{EI_n}$ ($10^{14}\,\mathrm{kg^{-1}}$) | n/a | Eqs. (1)–(4) | ✓ | × | 13.7 |
| Sensitivity: contrail model parameters (2019) | | | | | |
| No radiative heating effects | Cirium | Eqs. (1)–(4) | × | × | 66.8 |
| Contrail–contrail overlapping | Cirium | Eqs. (1)–(4) | × TS5 | ✓ | 59.1 |

that are up to 4 orders of magnitude larger than their alternative engine options (Teoh et al., 2024), which leads to a larger 2019 global mean nvPM $EI_n$ (1.39 vs. $1.02 \times 10^{15}$ $kg^{-1}$ in the baseline simulation, $+36\%$) and contrail net RF (Fig. 10c).

The simulation with a constant nvPM $EI_n$ of $10^{15}$ $kg^{-1}$ for all waypoints leads to a global contrail net RF of 74.8 $mW\,m^{-2}$ ($+20\%$ relative to the baseline). Regionally, this approach could underestimate the contrail net RF over Europe and the USA (Fig. 10d), likely because a higher proportion of flights are short-haul and utilise the Airbus A320 family, where the nvPM $EI_n$ from one of the engine options ($3–7 \times 10^{15}$ $kg^{-1}$) is significantly larger than the assumed $10^{15}$ $kg^{-1}$ (EASA, 2021; Teoh et al., 2024). A change in the assumed nvPM $EI_n$ from $10^{15}$ to $10^{14}$ $kg^{-1}$ leads to a smaller mean $\tau_{contrail}$ ($-50\%$), lifetime ($-35\%$), and coverage area ($-70\%$), which in turn produces a global contrail net RF of 13.7 $mW\,m^{-2}$ ($-82\%$) (Table S11). The simulation with an $EI_n$ of $10^{14}$ $kg^{-1}$ demonstrates the potential of fleet-wide reductions in engine particle emissions by an order of magnitude, but it is not included in our range of contrail RF estimates for 2019.

### 3.2.3   Contrail model parameters

The global contrail net RF from the baseline simulation (with radiative heating effects) is 8 % smaller than the simulation without radiative heating effects (62.1 vs. 66.8 $mW\,m^{-2}$). This is because solar and terrestrial radiation heats up the contrail plume and reduces the mean contrail lifetime by 23 % (2.4 vs. 3.0 h without radiative heating), which in turn lowers the contrail net RF over (i) regions with a higher fraction of aged contrails (Fig. S20) and (ii) Europe as less contrails are advected into the region via the North Atlantic jet stream (Fig. 10e). However, radiative heating also increases the vertical mixing rate and $\tau_{contrail}$ and causes the contrail net RF to be larger along established flight corridors (Fig. 10e). Seasonally, differences in the mean contrail net RF are largest in spring and summer (47.7 vs. 53.1 $mW\,m^{-2}$ without radiative heating, $-11\%$) and smallest in wintertime (84.0 vs. 87.3 $mW\,m^{-2}$, $-3.9\%$) (Fig. S17), because the impacts of radiative heating on the contrail lifetime depends on the magnitude of SDR and OLR.

The effects of contrail–contrail overlapping reduces the global annual mean contrail net RF by 5 % (from 62.1 $mW\,m^{-2}$ in the baseline to 59.1 $mW\,m^{-2}$), which is consistent with an earlier study that estimated a 3 % reduction in the global contrail net RF due to contrail–contrail overlapping (Sanz-Morère et al., 2021). Regionally, the change in contrail climate forcing depends on the magnitude of the annual mean contrail net RF, OLR, and the ratio of SW-to-LW RF (Fig. S21). The largest reduction in contrail climate forcing occurs in regions with dense air traffic, such as the USA ($-9.7\%$) and Europe ($-9.4\%$), while the contrail net RF is increased in areas with a higher prevalence of strongly cooling contrails, such as the eastern North Atlantic and northern Asia (Fig. 9b), because the change in SW RF is larger than the LW RF (Fig. 10f and Table S12).

### 3.3   Comparison with other studies

### 3.3.1   Inter-model comparisons

Lee et al. (2021) used the 2006 global annual mean contrail cirrus net RF estimates from three different global contrail models (Sect. S5) and extrapolated the contrail cirrus net RF to 2018 levels (111 [33, 189] $mW\,m^{-2}$), assuming that the growth in contrail climate forcing is proportional to the growth in global annual flight distance flown. Gettelman et al. (2021) applied a similar approach where they scaled the 2006 global air traffic to 2020 levels, assuming that the global air traffic distribution remains unchanged, and estimated a 2020 global annual mean contrail cirrus net ERF of $62 \pm 59$ $mW\,m^{-2}$ ($2\sigma$) in the absence of any COVID-19 disruptions. Our nominal 2019 global contrail cirrus net RF and ERF estimates are 44 % lower than the central RF estimate from Lee et al. (2021) (62.1 vs. 111 $mW\,m^{-2}$) and 58 % lower than the mean ERF estimate from Gettelman et al. (2021) (26.1 vs. 62 $mW\,m^{-2}$), and part of these discrepancies is due to a higher air traffic growth rate in the subtropics ($+12\%$ $yr^{-1}$ in China and India vs. $+6\%$ $yr^{-1}$ globally) between 2006 and 2018 (World Bank, 2023), where persistent contrails are less likely to form (Fig. 5). Differences in the contrail modelling approaches (i.e. general circulation models vs. Lagrangian models), model inputs, and parameter settings are also likely to contribute to discrepancies in the contrail climate forcing estimates between studies.

The global annual $EF_{contrail}$ and $CO_2$ emissions derived from this study are also used to estimate the 2019–2021 annual mean contrail cirrus $GWP_{20}$ and $GWP_{100}$ (cf. Eqs. 11 and 12 and Table 1). Our estimates (1.06 [TS6] for $GWP_{20}$ and 0.29 for $GWP_{100}$) are 54 % smaller than Lee et al. (2021) (2.32 for $GWP_{20}$ and 0.63 for $GWP_{100}$), owing to our lower relative contrail cirrus RF for the reasons discussed above. We note that the 2019–2021 annual $CO_2$ emissions (cf. Table 1) used to calculate the contrail GWPs in this study are 0.8 %–0.9 % lower than those provided by GAIA (Teoh et al., 2024), because the flight waypoints that were flown above 13 km were not included in the contrail simulation (Sect. 2.3), and our estimate of the 2019 annual $CO_2$ emissions (885 Tg) is ~ 14 % lower than that of Lee et al. (2021) (1034 Tg in 2018) due to differences between top-down and bottom-up estimates.

Recently, Bier and Burkhardt (2022) improved parameterisations of the initial contrail ice nucleation and particle losses in the ECHAM general circulation model and lowered their 2006 global contrail net RF from 56 $mW\,m^{-2}$ (Bock and Burkhardt, 2016a) to 43.7 $mW\,m^{-2}$. Our 2019 global annual mean contrail coverage area (0.66 % under clear-sky conditions) and net RF (62.1 $mW\,m^{-2}$) exceed their 2006 es-

timates (0.6 % and 43.7 mW m$^{-2}$) by 10 % and 42 %, respectively. The derived 2006–2019 average annual growth rate of the global contrail coverage area (+0.7 % yr$^{-1}$) and net RF (+2.7 % yr$^{-1}$) is lower than the growth in flight distance flown during the same period (+3.6 % yr$^{-1}$), and this could be explained by (i) the higher share of air traffic growth in the subtropics where $p_{\text{contrail}}$ is smaller than the global average (Fig. 5) and (ii) improvements in aircraft engine technology which reduced the fuel consumption per distance travelled by $\sim$ 6 % (from 4.87 kg km$^{-1}$ in 2006 to 4.60 kg km$^{-1}$ in 2019) and, in turn, is expected to lower the nvPM emissions per flight distance flown (cf. Eq. 5), contrail lifetime and coverage area, and the EF$_{\text{contrail}}$ per flight distance flown (Teoh et al., 2022).

Regionally, Teoh et al. (2022) used CoCiP without radiative heating effects and estimated a 2019 annual mean contrail net RF over the North Atlantic (235 mW m$^{-2}$) that is 22 % smaller than this study's estimates (300 mW m$^{-2}$). The contrail net RF values from Teoh et al. (2022) are likely underestimated because (i) their air traffic dataset only recorded flights that enter the Shanwick Oceanic flight information region (10–40° W), thereby capturing $\sim$ 80 % of the North Atlantic oceanic traffic (Molloy et al., 2022), and (ii) contrails that were formed outside the North Atlantic and subsequently advected into the domain were not accounted for.

### 3.3.2 Satellite observations and measurements

Several studies used satellite observations to estimate the global/regional contrail net RF and coverage area. Quaas et al. (2021) used satellite observations to compare the cirrus coverage before (2011–2019) and during the COVID-19 period (Spring 2020), where their estimated pre-COVID global mean contrail net RF (61 ± 39 mW m$^{-2}$) is within 1.8 % of our 2019 global annual mean contrail net RF (62.1 mW m$^{-2}$). Meijer et al. (2022) used geostationary satellite imagery and a machine learning algorithm to estimate the 2018–2019 annual mean contrail cirrus cover over the United States (0.15 %), which is around 50 % smaller than our 2019 estimates (0.31 %, Table 2). These differences may be due to the reduced probability for satellites detecting (i) freshly formed contrails with sub-pixel width, (ii) aged contrails that have lost their line-shaped structure, (iii) $\tau_{\text{contrail}} < 0.1$, and (iv) contrails that overlap with natural cirrus (Mannstein et al., 2010; Vázquez-Navarro et al., 2015). Nonetheless, when considering seasonal trends in the global contrail coverage area, our study aligns with findings from Stuber and Forster (2007), who calibrated their estimates with satellite observations, showing that the contrail coverage peaks in the spring and autumn and is at a minimum in the summer. Figure 11 also shows that our simulated contrail properties from CoCiP are generally in good agreement with in situ, remote sensing, and satellite observations from the contrail library database (COLI) (Schumann et al., 2017).

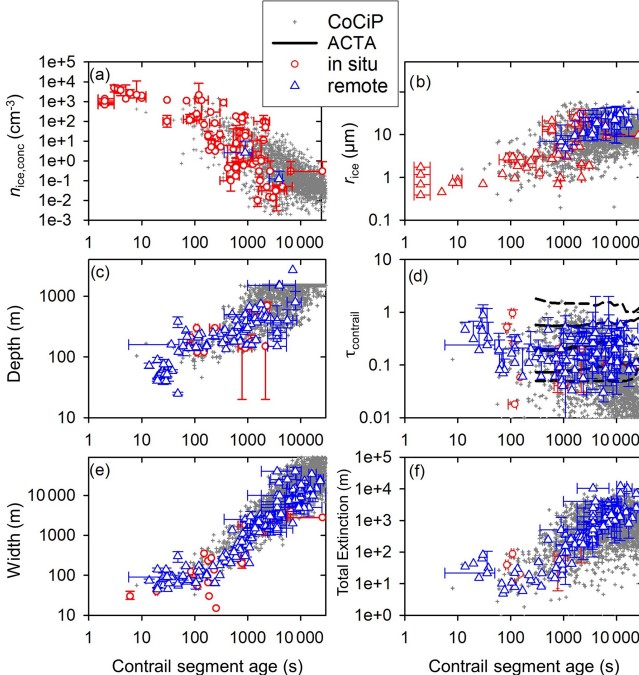

**Figure 11.** Comparison of the simulated contrail properties from CoCiP with in situ, remote sensing, and satellite observations from the contrail library database (COLI) (Schumann et al., 2017) versus the contrail segment age. The contrail properties compared include the contrail **(a)** ice particle number concentration in the plume ($n_{\text{ice,conc}}$); **(b)** $r_{\text{ice}}$; **(c)** depth; **(d)** $\tau_{\text{contrail}}$; **(e)** width; and **(f)** total extinction, i.e. the integral of the optical extinction over the contrail cross-sectional area, which influences the contrail RF'. The red data points are from in situ measurements, blue data points are from remote sensing, and the black lines in **(d)** represent the 0th, 10th, 50th, 90th, and 100th percentiles from the Meteosat Second Generation (MSG) satellites and the automatic contrail tracking algorithm (ACTA) (Vázquez-Navarro et al., 2015). The grey data points are the simulated CoCiP contrail properties from a subset of results within 06:00–09:00 UTC in 1 January and 15 July 2021.

## 4 Conclusions

We simulated the global contrail climate forcing for 2019–2021 using historical flight trajectories derived from ADS-B telemetry, improved nvPM EI$_n$ that accounts for specific aircraft–engine types, the ERA5 HRES reanalysis with global corrections applied to the humidity fields, and CoCiP. Table 3 summarises the global annual mean contrail net RF from the different simulation runs performed in this study.

Our 2019 global annual mean contrail cirrus net RF estimate (62.1 mW m$^{-2}$) is (i) 44 % lower than the central estimate from Lee et al. (2021), where the 2006 global contrail net RF from three studies were extrapolated to 2018 levels (111 [33, 189] mW m$^{-2}$), assuming the global growth in flight distance flown and contrail net RF are proportional, and (ii) 43 % higher than the 2006 estimate from Bier and Burkhardt (2022) (43.7 mW m$^{-2}$), where the derived 2006–

2019 average annual growth rate of the global contrail net RF ($+2.7\,\%\,\mathrm{yr}^{-1}$) is lower than the growth in flight distance flown ($+3.6\,\%\,\mathrm{yr}^{-1}$). These discrepancies are likely caused by the higher relative air traffic growth rate in the subtropics where persistent contrails are less likely to form because aircraft fly at lower altitudes and the Hadley Circulation limits the ISSR coverage area. Regionally, we estimate that Europe ($876\,\mathrm{mW\,m}^{-2}$), the USA ($414\,\mathrm{mW\,m}^{-2}$) and the North Atlantic ($300\,\mathrm{mW\,m}^{-2}$) have the largest contrail net RF in 2019, while the forcing in East Asia ($63.9\,\mathrm{mW\,m}^{-2}$) and China ($62.3\,\mathrm{mW\,m}^{-2}$) are close to the global mean value. Policy response to COVID-19 impacted global aviation operations and fleet composition, lowering the 2020 and 2021 annual mean contrail net RF to 27.3 and $31.7\,\mathrm{mW\,m}^{-2}$, respectively. Globally, the 2019–2021 annual mean contrail cirrus $\mathrm{GWP}_{100}$ and $\mathrm{GWP}_{20}$ are estimated to be 0.29 and 1.06, respectively (Table 1).

Around 20 % of all flights formed persistent contrails in 2019–2021 (i.e. contrails that survive the wake vortex phase), of which (i) 70 % of the contrail-forming flights have a warming effect ($\mathrm{EF}_{contrail} > 0$) and (ii) 10 % of the contrail-forming flights (or 2 % of all flights) were responsible for 80 % of the global annual $\mathrm{EF}_{contrail}$ (Fig. 8). The most strongly warming and cooling contrail segments are generally formed (i) at higher latitudes and in specific regions (i.e. North Atlantic and Southeast Asia), depending on the spatiotemporal variations in meteorology, radiation, and tropopause height; (ii) in specific origin–destination routes that are related to flight scheduling factors; and (iii) by aircraft–engine types with high nvPM number emissions that are larger than the mean fleet-aggregated values (Fig. 9, Tables S9 and S10). These results indicate the potential to significantly reduce the contrail climate forcing by targeting the mitigating actions to a small subset of flights and regions associated with strongly warming contrails.

The 2019 global annual mean contrail net RF is most sensitive to the humidity corrections applied to the ERA5 HRES ($34.8\,\mathrm{mW\,m}^{-2}$ without humidity correction, $-44\,\%$ relative to the baseline simulation), followed by assumptions on the aircraft–engine assignment and nvPM emissions (73.1–$74.8\,\mathrm{mW\,m}^{-2}$, $+18\,\%$–20 %), and it is least sensitive to the radiative heating ($66.8\,\mathrm{mW\,m}^{-2}$, $+7.6\,\%$) and contrail–contrail overlapping effects ($59.1\,\mathrm{mW\,m}^{-2}$, $-4.8\,\%$) (Table 3). The combined results of our sensitivity analysis suggest that the 2019 global annual mean contrail net RF could range between 34.8 and $74.8\,\mathrm{mW\,m}^{-2}$.

Future work should be prioritised towards (i) performing inter-model comparison studies to understand differences in the estimated contrail climate forcing from CoCiP versus those provided by other contrail models (Chen and Gettelman, 2013; Bier and Burkhardt, 2022; Fritz et al., 2020); (ii) evaluating the overall uncertainty in the simulated contrail climate forcing by propagating uncertainties in various input parameters, including meteorology, aircraft–engine performance and particle number emissions, ice particle habits, and radiation transfer scheme; (iii) systematically comparing simulated contrail properties with in situ measurements and observations from ground-based cameras, lidar, and satellites; (iv) evaluating the ERF / RF ratio at regional scale and the effects of contrail cirrus climate forcing on surface temperature change; and (v) advancing mitigation of contrail climate forcing via forecasting and flight trajectory optimisation, alternative fuels, and cleaner-burning engines.

**Code and data availability.** The Global Aviation emissions Inventory based on ADS-B (GAIA) dataset (Teoh et al., 2024) is provided as gridded outputs on Zenodo for the full year of 2019 (https://doi.org/10.5281/zenodo.8369829, Teoh et al., 2023a) and bi-monthly for 2020 and 2021 (https://doi.org/10.5281/zenodo.8369925, Teoh et al., 2023b). Flight trajectory and aircraft fuel consumption data are commercially sensitive, and the flight-waypoint and flight-summary outputs can be made available for scientific research upon reasonable request. The pycontrails repository that contains the CoCiP algorithm has recently been published and publicly available at https://doi.org/10.5281/zenodo.7776686 (Shapiro et al., 2023). IAGOS data were created with support from the European Commission; national agencies in Germany (BMBF), France (MESR), and the UK (NERC); and the IAGOS member institutions (https://www.iagos.org/organisation/members/, last access: 2 May 2024). The participating airlines (Deutsche Lufthansa, Air France, Australian Airlines, China Airlines, Iberia, Cathay Pacific, Air Namibia, Sabena) have supported IAGOS by carrying the measurement equipment free of charge since 1994. The data are available at https://doi.org/10.25326/06 (Boulanger et al., 2022).

**Supplement.** The supplement related to this article is available online at: https://doi.org/10.5194/acp-24-1-2024-supplement.

**Author contributions.** RT, US, CV, and MEJS conceptualised the study, developed the methodology, and undertook the investigation. RT, ZE, US, SR, and MS were responsible for software development and data curation. RT, MEJS, and US created or sourced the figures. RT wrote the original manuscript. RT, ZE, US, CV, MS, and MEJS reviewed and edited the manuscript. MEJS, MS, and CV acquired funding. All authors have read, edited, and reviewed the manuscript, and they agreed upon the published version of the paper.

**Competing interests.** The contact author has declared that none of the authors has any competing interests.

**Disclaimer.** This document used elements of Base of Aircraft Data (BADA) Family 4 Release 4.2, which has been made available by EUROCONTROL to Imperial College London. EUROCONTROL has all relevant rights to BADA. © 2019 The European Organisation for the Safety of Air Navigation (EUROCONTROL).

EUROCONTROL shall not be liable for any direct, indirect, incidental, or consequential damages arising out of or in connection with this document, including the use of BADA. This document contains Copernicus Climate Change Service information 2023. Neither the European Commission nor ECMWF is responsible for any use of the Copernicus information.

**Acknowledgements.** The computational resources required to perform the global contrail simulations are supported by Google through cloud credits on the Google Cloud Platform. We thank Andreas Petzold for reviewing an earlier version of the paper.

**Financial support.** Roger Teoh and Marc E. J. Stettler are supported by Innovate UK (grant no. 10049507). Christiane Voigt has been supported by Deutsche Forschungsgemeinschaft (DFG) under contract nos. VO 1504/7-1 and VO 1504/9-1 and by the EU Horizon 2020 under the Advancing the Science for Aviation and Climate (ACACIA) project (grant no. 875036).

**Review statement.** This paper was edited by Yuan Wang and reviewed by one anonymous referee.

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

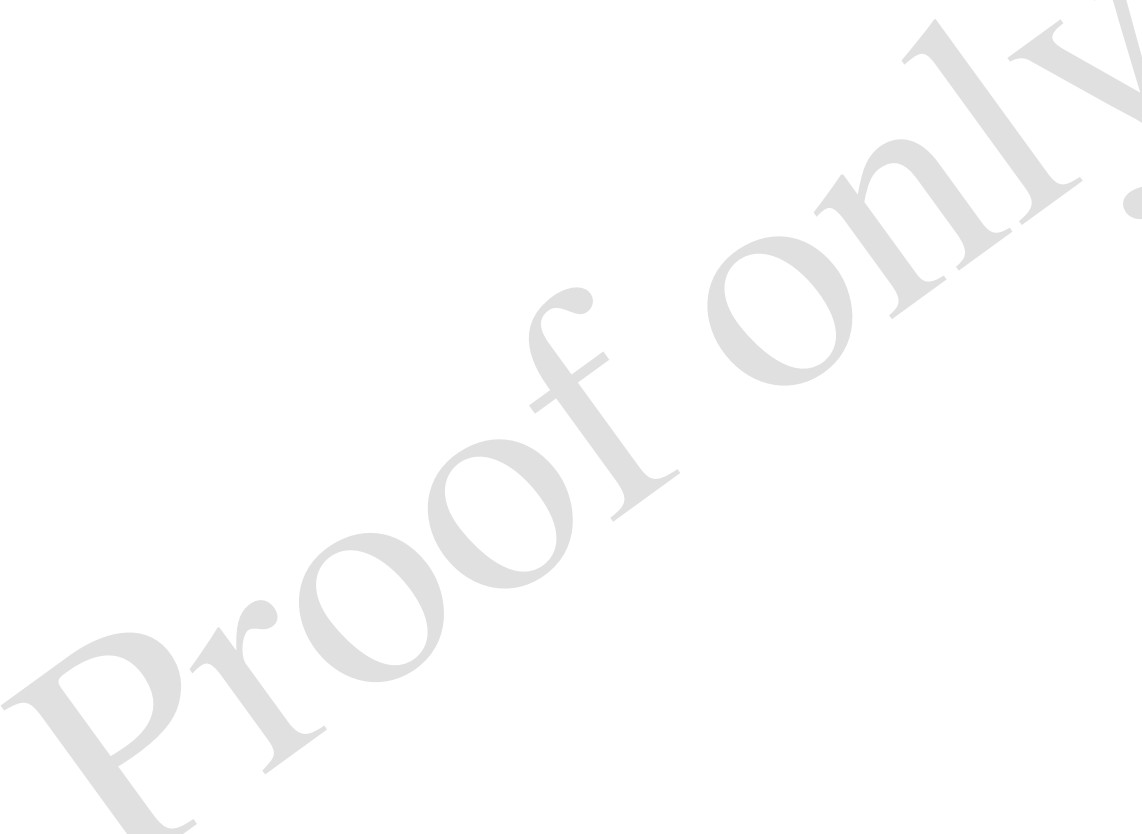

## Remarks from the language copy-editor

CE1    Please confirm the adjustments to align with our standards: comma removed as it's not a full clause.

CE2    Please confirm the adjustments to align with our standards: colon removed as a full clause must precede a colon.

CE3    Please confirm the changes.

CE4    Please confirm.

## Remarks from the typesetter

TS1    Please confirm.

TS2    Due to the requested changes, we have to forward your requests to the handling editor for approval. To explain the corrections needed to the editor, please send me the reason why these corrections are necessary. Please note that the status of your paper will be changed to "Post-review adjustments" until the editor has made their decision. We will keep you informed via email.

TS3    Please confirm.

TS4    Please confirm.

TS5    Due to the requested changes, we have to forward your requests to the handling editor for approval. To explain the corrections needed to the editor, please send me the reason why these corrections are necessary. Please note that the status of your paper will be changed to "Post-review adjustments" until the editor has made their decision. We will keep you informed via email.

TS6    Due to the requested changes, we have to forward your requests to the handling editor for approval. To explain the corrections needed to the editor, please send me the reason why these corrections are necessary. Please note that the status of your paper will be changed to "Post-review adjustments" until the editor has made their decision. We will keep you informed via email.

TS7    Due to the requested changes, we have to forward your requests to the handling editor for approval. To explain the corrections needed to the editor, please send me the reason why these corrections are necessary. Please note that the status of your paper will be changed to "Post-review adjustments" until the editor has made their decision. We will keep you informed via email.

TS8    This data set is not listed in the data availability section. Please check.

TS9    This data set is not listed in the data availability section. Please check.

TS10    This data set is not listed in the data availability section. Please check.

TS11    This data set is not listed in the data availability section. Please check.

TS12    Please provide date of last access.