# Peer review of "Global aviation contrail climate effects from 2019 to 2021"

_EGUsphere, 2023_

## Author Response (AR1)

**Response to Reviewer Comments**

We express our gratitude to the referee and two non-referees for their insightful comments, which significantly contributed to enhancing the quality and clarity of this manuscript. The *italicized text* below reflects the reviewer's remarks, while our responses are presented in normal text. Blue text is used to cite from the revised manuscript. When page and line numbers are specified, they refer to the clean version of the revised manuscript.

**REFEREE 1 (RC1)**

**General comments**

*This is a mostly well-written paper on the estimation of global contrail climate effects for 2019 to 2021. This study has implemented almost all the most-up-to-date data and methods related to air traffic, aviation particle emission, meteorological background, and contrail formation, etc. This study explores the global contrail properties and climate forcing for 2019–2021; identifies the set of conditions that causes strongly warming/cooling contrails; evaluates the sensitivity of the simulated contrail climate forcing to aircraft emissions, meteorology, and contrail model parameters; and compares their new global contrail RF estimates with existing studies. The results look valid and the sensitivity experiments are reasonably set to illustrate the different factors. I have only a few suggestions to improve the paper for the authors' consideration.*

**Major concerns**

1. *For the methodology part, I suggest the authors added a flowchart and the related descriptions to better explain how the contrails are simulated as well as how the radiative forcings are calculated.*

   - Thank you for this suggestion. We have added a flow chart at the start of the methodology section to summarise the different datasets and models that were used in this study:

[Figure]

**Figure 1: Flowchart summarising the dataset, models, and input parameters that are used in this study.**

- We have also revised the manuscript to incorporate these suggestions:
    - [Main text: Line 93] "**Fig. 1 summarises the datasets, models, and input parameters that are used in this study.**"
    - [Main text: Lines 181 – 183] "**The local contrail RF (RF') for each contrail segment, i.e., the change in radiative flux over the area covered by the contrail, is estimated using a parametric RF model that was developed by Schumann et al., wherein the simulated contrail properties from CoCiP and meteorology from the ERA5 HRES are served as inputs.**"

2. *Line 285: It is interesting to find the largest interannual variability in p-contrail at high latitudes. But it is not very convincing to attribute the reason to small sample size. Could the authors explain more about it? Moreover, if the sample size is a critical issue, it is worthwhile to show and discuss about it somehow.*

- Several factors collectively contribute to the large inter-annual variability in the percentage of flight distance forming persistent contrails ($p_{contrail}$) at high latitudes:

    i. The smaller grid cell area at high latitudes, because Earth's surface area decreases as one moves towards higher latitudes which ultimately converges to a singular point at the poles, which likely contributes to a larger relative inter-annual variability in the coverage area of ice-supersaturated regions (ISSR), see figure attached at the end of this point, and

    ii. The low air traffic activity at high latitudes, where 0.62% and 0.06% of the global annual flight distance in 2019 were flown at latitudes above 66.5°N and below 45°S respectively (Teoh et al., 2024), where $p_{contrail}$ at these latitude bins are calculated from a significantly smaller sample size relative to other latitudes.

- We have made the following changes in the revised manuscript and Supplement to address this point:
    - [Main text: Lines 325 – 331] "Figure 5: The percentage of annual flight distance flown that formed persistent contrails ($p_{contrail}$) by latitude in 2019 (blue line), 2020 (orange line), and 2021 (green line). **Several factors collectively contribute to** the large inter-annual variability in $p_{contrail}$ at high latitudes (above 60°N and below 60°S)**, including the: (i) smaller domain area at high latitudes, which can cause a larger inter-annual variability in the ISSR occurrence relative to other latitude bands (see Fig. S14); and (ii) low air traffic activity at high latitudes where 0.62% and 0.06%** of the global annual flight distance were flown a**t** latitudes **above 66.5°N and below 45°S respectively (Teoh et al., 2024), thereby causing $p_{contrail}$ at these latitude bins to be calculated from a significantly smaller sample size relative to other latitudes.**"

    - [Supplement: Lines 368 – 376] "**Fig. S14 shows the monthly-averaged ISSR occurrence at different latitude bins, represented as a percentage of the airspace volume. Notably, the ISSR occurrence at high latitudes (75°N - 90°N) exhibits a larger inter-annual variability relative to other latitude bands, likely due to its smaller grid cell area. Additionally, the ISSR occurrence between December 2019 and April 2020 is also around two times larger than the 2019–2021 annual averages. These factors, coupled with the**

**low air traffic activity at high latitudes (0.62% and 0.06% of the global annual flight distance were flown above 66.5°N and below 45°S respectively), likely contributed to the large inter-annual variability in $p_{contrail}$ between 2019 and 2021, as presented in Fig. 5 in the main text."**

o [Supplement, pages 381 – 384]

[Figure]

**Figure S2: Monthly average ISSR coverage, expressed as a percentage of airspace volume, from 2019 to 2021 across at different latitude bands: 0 – 30°N (blue line), 30°N – 60°N (orange line), 60°N – 90°N (green line), and 75°N – 90°N (red line).**

3. *Could the authors show some comparisons between the simulated and observed contrails? I know this could be hard and difficult, but some more evidence help increase the credibility.*

   - We acknowledge the significance of conducting further comparisons between the simulated and observed contrails. To the best of our ability, we have compared the simulated contrail properties with observations and measurements compiled from the contrail library database (COLI). See Fig. 11 in the revised manuscript.

   - We note that further work is currently ongoing to compare the simulated contrail properties from CoCiP with satellite observations and ground-based cameras. These comparisons will be submitted in separate manuscripts that are currently in preparation.

   - We encourage future studies to provide observational evidence on contrails in different regions.

**Minor problems**

4. *Line 14: It is better to explain the meanings of the numbers in square brackets (111 [33, 189] mW m⁻²).*

   - Thank you for highlighting this. The squared brackets denote the 95% confidence interval. However, we have identified inconsistencies across the manuscript where square brackets were incorrectly used to describe the results of our sensitivity analysis, which did not represent the 95% confidence interval.

   - We have made the following changes throughout the manuscript to address this point:

- o [Main text: Lines 13 – 14] "Our 2019 global annual mean contrail net RF (62.1 mW m$^{-2}$) is 44% lower than current best estimates for 2018 (111 [33, 189] mW m$^{-2}$, **95% confidence interval**)."
- o [Main text: Lines 24 – 25] " **Using this** sensitivity analysis, we estimate **that the** 2019 global **annual mean** contrail net RF  **could range between 34.8 and 74.8**  mW m$^{-2}$."
- o [Main text: Lines 447 – 448] "**When taken together, the sensitivity analysis estimates a 2019 global annual mean contrail cirrus net RF that falls within the range of 34.8 and 74.8 mW m$^{-2}$.**"
- o [Main text: Lines 588 – 589] " **The combined results of** our sensitivity analysis  **suggest that the** 2019 global **annual mean** contrail net RF **could** range **between**  and 74.8 mW m$^{-2}$."

5. *Line 15: It is desirable to plot the boxes of the areas of US, Europe, and especially East Asia on the global maps (i.e., Fig.1) for better understanding of the results.*
   - Thank you for this suggestion. We have relocated the plot of the regional spatial bounding boxes from the Supplement to the main text (Fig. 2 in the revised manuscript) and made the following changes to address this point:
     - o [Main text: Lines 200 – 203]

[Figure]

Figure 2: Spatial bounding box used to estimate the regional air traffic, emissions, and contrail properties. The specific dimensions of these bounding boxes can be found in Table S5 in the Supplement. Basemap plotted using Cartopy 0.21.1 © Natural Earth; license: public domain.

   - o [Main text: Lines 196 – 198] "The regional contrail properties and climate forcing are estimated using rectangular spatial bounding boxes (**Fig. 2 and** Table S5 ) that are consistent with previous studies (Wilkerson et al., 2010; Hoare, 2014; Teoh et al., 2024)."

6. *Line 80: It is suggested to note which section corresponds to the four objectives.*

   - The following changes are made in the main text to cross-reference the stated research objectives to the relevant sections in the manuscript:

o [Main text: Lines 80 – 84] "In this study, we use a new global aviation emissions inventory based on Automatic Dependent Surveillance–Broadcast (ADS-B) telemetry (GAIA) (Teoh et al., 2024) to: (i) quantify the global contrail properties and climate forcing for 2019–2021 **(Section 3.1)**; (ii) identify the set of conditions that causes flights to form strongly warming/cooling contrails **(Section 3.1.4)**; (iii) evaluate the sensitivity of the simulated contrail climate forcing to aircraft emissions, meteorology, and contrail model parameters **(Section 3.2)**; and (iv) compare our global contrail RF estimates with existing studies **(Section 3.3)**."

7. *Line 91: It seems better to list the materials included in the SI instead of stating "Further information not included in the main text can be found in the SI." I suppose you can't include everything.*

   • Thank you for this suggestion. We have made the following changes to address this point:

   o [Main text: Lines 93 – 95] "Further **methodological** information **on the: (i) formulation of the extended humidity correction model; (ii) various output formats provided by CoCiP; and (iii) approach to simulate the effects of contrail-contrail overlapping is described in detail**  in the  **Supplement**."

   o [Main text: Lines 264 – 265] "**Additional data tables and statistics from the global contrail simulation that are not presented here can be found in the Supplement as referenced in the text.**"

8. *Line 202: Why and what can be the potential impacts by assuming an ERF/RF ratio of 0.42? The authors may need to explain a bit.*

   • Thank you. We agree with this suggestion and have now addressed the following points in the revised manuscript:

   i.   defined the effective radiative forcing (ERF) metric,

   ii.  discussed the difference between the radiative forcing (RF), ERF metrics, and the contrail efficacy,

   iii. more clearly highlight the limitations of CoCiP which do not directly provide the ERF estimates, and

   iv.  Emphasised that the ERF/RF ratio is only used at a global scale, rather than on individual flights, in order to compare the global annual mean contrail climate forcing with existing studies.

   • The following change has been made to the main text to address this point:
   o [Main text: Lines 217 – 227] " **Next,** global annual mean contrail  is estimated **from the RF** by assuming a **mean** ERF/RF ratio of 0.42 (Lee et al., 2021). **The ERF accounts for the rapid atmospheric adjustments (i.e., atmosphere-humidity exchange and temperature lapse rate) and natural cirrus responses (i.e., reduction in natural cirrus occurrence and cloudiness) resulting from the contrail** (Lee et al., 2023)**. Thus, the ERF/RF ratio is a measure of efficacy which describes how effective the contrail RF impacts the global mean surface temperature**

compared to the $CO_2$-induced RF (Myhre et al., 2013). **Our assumed ERF/RF ratio (= 0.42) is based on three global climate model studies that estimate the ERF to range between 0.31 and 0.59** (Ponater et al., 2005; Rap et al., 2010; Bickel et al., 2019), **although a lower ERF/RF ratio of 0.21 was estimated from a recent coupled atmosphere-ocean climate model** (Bickel, 2023). **Due to the large uncertainty and spatiotemporal variabilities in the contrail efficacy** (Ponater et al., 2005; Schumann and Mayer, 2017; Gettelman et al., 2021), **we: (i) base our analysis on the instantaneous contrail climate forcing (RF and EF$_{contrail}$); and (ii) only apply the ERF/RF conversion at a global scale, rather than on individual flights, focusing solely on comparing our global annual mean contrail ERF with existing studies** (Lee et al., 2023)."

9. *Line 221: It could be easier for the readers to understand the points of the sensitivity experiments if more explanations be given on why such settings were used.*

- Thank you for this suggestion. We have made the following changes to the main text to address this point:

  o [Main text: Lines 36 – 39] "The nvPM acts as the primary source of condensation nuclei in the "soot-rich" regime, defined when the soot number emissions index (EI$_n$) exceeds **a threshold of around** $10^{14\cancel{13}}$ kg$^{-1}$, while ambient aerosols, organic and sulfuric particles can nucleate under "soot-poor" conditions (EI$_n$ < $10^{14\cancel{13}}$ kg$^{-1}$) (Kärcher and Yu, 2009; Kärcher, 2018)."

  o [Main text: Lines 243 – 258] "**To assess the sensitivity of CoCiP to various inputs and contrail model parameters,** we perform a sensitivity analysis by re-running the global contrail simulation for 2019 with seven distinct set-ups: (i) a simulation without humidity corrections applied to the ERA5 HRES (Section 2.2); (ii) a simulation using a constant humidity correction that was adopted in earlier studies (Schumann, 2012; Schumann et al., 2015; Teoh et al., 2020; Schumann et al., 2021), where the ERA5-derived RHi fields were uniformly increased by dividing it with a factor of 0.95; (iii) a simulation that uses the default aircraft-engine combination from BADA3 (EUROCONTROL, 2019), instead of the specific aircraft variant and engine model provided by a fleet database (Cirium, 2022);  two simulations where **all waypoints are assumed with** a constant nvPM EI$_n$ of **(iv)** $10^{15}$ **kg$^{-1}$; and (v)** $10^{14}$ kg$^{-1}$ **respectively**; (v**i**) a simulation without the effects of radiative heating interactions with the contrail plume; and (v**ii**) a simulation that approximates the change in contrail climate forcing due to contrail-contrail overlapping (methodology detailed in the **Supplement Sect.** S4.3).

  **Sensitivity experiments (i), (iii), (vi), and (vii) are set up to assess the impact of improved input parameters and updates to the contrail modelling processes on the simulated contrail climate forcing (Schumann et al., 2021b, 2010), while sensitivity experiments (ii), (iii), and (iv) are designed to align with the methodology of previous studies and explore their potential implications (Schumann, 2012b; Schumann et al., 2015b; Teoh et al., 2020b; Bier and Burkhardt, 2022b). In sensitivity experiment (v), the nvPM EI$_n$ is fixed at the threshold marking the transition from 'soot-rich' to 'soot-poor' conditions (~$10^{14}$ kg$^{-1}$) to estimate the minimum contrail climate forcing that could be achieved through reductions in aircraft nvPM emissions.**"

**NON-REFEREE COMMENT 1: Louis Meuric (CC1 – CC3)**

10. *It would be very useful to calculate and disseminate the ERF for each country of the northern hemisphere: there are still obvious differences (Spain / Norway)*

- Thank you for this suggestion. Although this can theoretically be done, we have previously considered this and decided against breaking down the contrail climate forcing to an individual country level for the following reasons:

  i. Unlike aircraft emissions, persistent contrails can form outside (inside) a specific country and subsequently advect into (out of) the country over time, and this phenomenon becomes increasingly significant for small countries which could distort these country-level statistics, and

  ii. Persistent contrails formed in a country's airspace may be caused by international overflights that are not related to the country's air traffic activity itself.

- To minimise these subjective interpretations, we have therefore decided to stick to the regional spatial bounding boxes that were defined in previous studies (see Fig. 2 in the revised manuscript).

11. *Moreover, a study carried out on the Tibetan plateau, where warming accelerated during the winter months at the end of the 20th century, shows that increased surface humidity leads to an increase in long-wave radiation (= heat) and can locally and temporarily raise temperatures at altitude. https://agupubs.onlinelibrary.wiley.com/doi/full/10.1029/2009GL037245. Is it possible to have induced ERF at an altitude between 1500 – 2500 m?*

- Thank you for highlighting this observation. We note that this observation might be less relevant for contrails for the following reasons:

  i. Air traffic activity is notably sparse over the Tibetan Plateau (Teoh et al., 2024), primarily due to safety concerns such as its high average terrain and increased turbulence risk. Consequently, the global annual mean net contrail radiative forcing (RF) in this region is very small ($< 10$ mW m$^{-2}$, see Fig. 3a in the revised manuscript), and

  ii. Persistent contrails generally form at altitudes between 9 km (30,000 feet) and 12 km (40,000 feet) and the contrail-induced heating tends to be at a maximum just below the contrail layers (Meerkötter et al., 1999). In contrast, the magnitude of Earth's surface warming induced by contrails is largely influenced by atmospheric mixing conditions, where it tends to have a: (1) larger relative impact when the atmospheric mixing is strong as a higher proportion of thermal heat can be advected to the surface; and (2) smaller relative impact when the atmosphere is stable because an increasing fraction of the thermal heat is radiated to space before it is transferred to the surface (Schumann and Mayer, 2017).

- There are several studies that used general circulation models to quantify the contrail-induced ERF at cruise altitudes (Bickel et al., 2019; Ponater et al., 2021; Rap et al., 2010; Bickel, 2023), but only a limited number of studies that propagate the contrail-induced heating to lower altitudes and/or the surface (Ponater et al., 2005; Gettelman et al., 2021). Therefore, it is essential for more studies to evaluate the spatiotemporal variability in the contrail ERF and efficacy (i.e., how effective the contrail RF impacts the global mean surface temperature compared to the $CO_2$-induced RF).

- We also note that a short discussion on the contrail RF, ERF, efficacy, and surface temperature response have now been added in Section 2.4 to address Point (8).

12. (Ponater et al., 2021; Bickel et al., 2019)*The monthly distribution of the contrail coverage is slightly different from the one reported for year 2002 in the publication below, from Stuber & Forster : https://acp.copernicus.org/articles/7/3153/2007/. page 4, but 2002 was a special year, just after september 11.*

*About the monthly distribution, please find another source : https://commons.wikimedia.org/wiki/File:Contrails_1994_1995_Nasa_US_Air_Force.jpg?uselang=fr. Both statistics show a rebound in october and a high level, rather in February-April. This is important because one can link it with the first snows in october and the melting in march-april.*

- Our 2019 global contrail cirrus coverage (0.05 – 0.07%, Fig. 7f) is around 10-50% larger than those reported in Stuber & Forster (2007) (0.024 – 0.062%). Our larger estimate can most likely be attributed to the growth in global air traffic activity between 2002 and 2019. Additionally, the source and spatiotemporal resolution of the meteorological datasets, and the methodologies used to estimate the global contrail cirrus coverage area, may also contribute to discrepancies between our study and Stuber & Forster (2007).

- A comparison of the seasonal trend in global contrail coverage area between our study and Stuber & Forster (2007) exhibited consistency, where the global contrail cirrus cover peaks in the spring and autumn and is at a minimum in the summer.

- We have revised Section 3.3 (Comparison with other studies) to compare our results with additional contrail studies, including those from Stuber & Forster (2007), to address this comment:

  o [Main text: Lines 501 – 508] "**Gettelman et al. (2021) applied a similar approach where they scaled the 2006 global air traffic to 2020 levels, assuming that the global air traffic distribution remains unchanged, and estimated a 2020 global annual mean contrail cirrus net ERF of 62 ± 59 mW m$^{-2}$ (2σ) in the absence of any COVID-19 disruptions.**  Our nominal 2019 global contrail cirrus net RF **and ERF**  **estimates are**  44% lower than the  central **RF** estimates from Lee et al. (2021) (**62.1 vs.** 111 mW m$^{-2}$) and 58% lower than the mean ERF estimate from Gettelman et al. (2021) (26.1 vs. 62 mW m$^{-2}$), and  **part of these discrepancies** is due to a higher air traffic growth rate in the subtropics (+12% per annum in China and India vs. +6% globally) between 2006 and 2018 (World Bank, 2023), where persistent contrails are less likely to form (Fig. **5**)."

  o [Main text: Lines 521 – 528] "Our 2019 global annual mean contrail **coverage area (0.66% under clear-sky conditions) and** net RF (62.1 mW m$^{-2}$) exceed  their 2006 estimates (**0.6% and** 43.7 mW m$^{-2}$) **by 10% and 42% respectively**. The derived **2006–2019** average annual growth rate of the global contrail **coverage area (+0.7% per annum) and** net RF  (+2.7% per annum) is lower than the growth in flight distance flown during the same period (+3.6% per annum), and could be explained by: (i) the higher share of air traffic growth in the subtropics where $p_{contrail}$ is smaller than the global average (Fig. 5); and (ii) improvements in aircraft engine technology which

reduced the fuel consumption per distance travelled by ~6% (from 4.87 kg km$^{-1}$ in 2006 to 4.60 kg km$^{-1}$ in 2019) and, in turn, is expected to lower the nvPM **emissions per flight distance flown** (c.f. Eq. (5)), **contrail lifetime and coverage area**, and **the** EF$_{contrail}$ per flight distance flown (Teoh et al., 2022)."

o [Main text: Lines 544 – 554] "**Several studies used satellite observations to estimate the global/regional contrail net RF and coverage area. Quaas et al.** (2021) **used satellite observations to compare the cirrus coverage before (2011 – 2019) and during the COVID-19 period (Spring 2020), where their estimated pre-COVID global mean contrail net RF (61 ± 39 mW m$^{-2}$) is within 1.8% of our 2019 global annual mean contrail net RF (62.1 mW m$^{-2}$).** Meijer et al. (2022) used geostationary satellite imagery and a machine learning algorithm to estimate the 2018–19 annual mean contrail cirrus cover over the United States (0.15%), which is around 50% smaller than our 2019 estimates (0.31%, Table 2). These differences may be due to the reduced probability for satellites detecting: (i) freshly formed contrails with sub-pixel width; (ii) aged contrails that have lost their line-shaped structure; (iii) $\tau_{contrail}$ < 0.1; and (iv) contrails that overlap with natural cirrus (Mannstein et al., 2010; Vázquez-Navarro et al., 2015). **Nonetheless, when considering seasonal trends in the global contrail coverage area, our study aligns with findings from Stuber & Forster** (2007)**, which calibrated their estimates with satellite observations, showing that the contrail coverage peaks in the spring and autumn and is at a minimum in the summer.**"

**NON-REFEREE COMMENT 2: Adam Durant (CC4)**

*This is a much needed bottom up flight by flight assessment of the global annual radiative impact from aircraft contrails. The work that has gone into the analysis is impressive. Congratulations to the authors.*

13. *One aspect that does require a little more explanation is the use of a global climate model water vapour dataset, adjusted through a statistical fit against IAGOS in situ data, to drive highly specific trajectory-based analyses of radiative forcing from COCIP.*

*The data presented in supplementary tables shows "ratio compares the false positive and false negative rate and is computed by (NIAGOS/YHRES (%) − 1). A positive value indicates YIAGOS/NHRES (%) that the ERA5 HRES underpredicts contrails, a value of zero indicates a symmetrical false positive and false negative rate, while a negative value indicates that the ERA5 HRES overpredicts contrails." On semantics, this metric predicts ISSR occurrence, not contrails. The FP and FN rates are hidden in the ratio. The ETS values appear quite low.*

• Thank you for this suggestion. We have combined Tables S1 and S3 for clarity improvements, and updated its caption and footnote to address the point that was raised above,

o [Supplement: Lines 129 – 143]

Table S1: Comparison of the ISSR occurrence **between** the **in-situ RHi measurements from the** IAGOS **campaign in 2019 versus the RHi derived from the ERA5 HRES: (a) without humidity correction; and (b) with the global humidity correction, c.f. Eq. (S6) to Eq. (S9)**.  Y$_{IAGOS}$ indicates that the waypoint has an RHi > 100% (ISSR occurrence) according to the **in-situ** measurements, while N$_{IAGOS}$ indicates the opposite. The subscript

"HRES" is used to indicate ISSR occurrence as provided by the ERA5 HRES. **For (b), the metrics that are highlighted in green (red) indicates that its performance has improved (degraded) relative to (a).**

| | No. of waypoints | $Y_{IAGOS}/Y_{HRES}$ (%) | $N_{IAGOS}/N_{HRES}$ (%) | $Y_{IAGOS}/N_{HRES}$ (%) | $N_{IAGOS}/Y_{HRES}$ (%) | Ratio[a] | CvM stat[b] | ETS[c] |
|---|---|---|---|---|---|---|---|---|
| **(a)   IAGOS vs. ERA5 HRES (No RHi correction)** | | | | | | | | |
| 0 - 10°N | 20650 | 9.16 | 70.1 | 8.22 | 12.5 | -0.341 | 58.2 | 0.207 |
| 10 - 20°N | 48366 | 5.02 | 83.2 | 5.67 | 6.06 | -0.064 | 73.0 | 0.246 |
| 20 - 30°N | 144910 | 2.90 | 90.1 | 2.90 | 4.08 | -0.290 | 43.7 | 0.264 |
| 30 - 40°N | 141131 | 4.42 | 87.7 | 3.69 | 4.14 | -0.110 | 93.1 | 0.322 |
| 40 - 50°N | 114018 | 5.40 | 85.1 | 6.24 | 3.31 | 0.889 | 261 | 0.315 |
| 50 - 60°N | 106993 | 6.75 | 83.1 | 6.39 | 3.73 | 0.714 | 232 | 0.347 |
| 60 - 90°N | 33762 | 5.57 | 87.0 | 5.06 | 2.33 | 1.169 | 91.7 | 0.390 |
| **(b)   IAGOS vs. ERA5 HRES (Global humidity correction)** | | | | | | | | |
| 0 - 10°N | 20650 | 7.82 | 71.8 | 9.56 | 10.9 | -0.119 | 2.09 | 0.183 |
| 10 - 20°N | 48366 | 4.44 | 84.1 | 6.25 | 5.21 | 0.199 | 2.55 | 0.229 |
| 20 - 30°N | 144910 | 2.58 | 90.7 | 3.22 | 3.50 | -0.080 | 9.93 | 0.249 |
| 30 - 40°N | 141131 | 4.28 | 88.0 | 3.83 | 3.87 | -0.010 | 24.2 | 0.319 |
| 40 - 50°N | 114018 | 6.70 | 83. 7 | 4.94 | 4.69 | 0.054 | 1.06 | 0.358 |
| 50 - 60°N | 106993 | 8.40 | 81.5 | 4.74 | 5.40 | -0.122 | 22.3 | 0.394 |
| 60 - 90°N | 33762 | 6.93 | 86.1 | 3.70 | 3.28 | 0.128 | 0.360 | 0.456 |

[a]: Ratio compares the false positive and false negative rate and is computed by ($\frac{YN_{IAGOS}/NY_{HRES} \, (\%)}{NY_{IAGOS}/YN_{HRES} \, (\%)} - 1$). A positive value indicates that the ERA5 HRES underpredicts **ISSR occurrence**, a value of zero indicates a symmetrical false positive and false negative rate, while a negative value indicates that the ERA5 HRES overpredicts **ISSR occurrence**.

[b]: CvM test statistic, where a lower value indicates a better goodness-of-fit between the probability density function of the measured and ERA5-derived RHi.

[c]: The equitable threat score (ETS) is calculated according to Appendix A of Gierens et al. (2020), where ETS = 1 indicates that the ERA5-derived RHi is in perfect agreement with measurements, ETS = 0 indicates a completely random relationship, while ETS < 0 indicates an inverse relationship between the measured and ERA5-derived RHi.

- In addition, we also updated the results in Table S3 and revised the discussion of these results. The performance metrics are now calculated using the weighted-mean of the number of waypoints in each bin, instead of the simple-mean:
  - [Supplement: Lines 165 – 176]:

Table S3: **Comparison of different** **p**erformance metrics **to evaluate** the agreement between the RHi measurements **provided by** the full IAGOS dataset **for 2019** versus the uncorrected and corrected ERA5 HRES global humidity fields. **These metrics are weighted by the number of waypoints in each latitude bin (see Table S1).**

| **Full IAGOS dataset vs. ERA5 HRES** | **Correct prediction (%)** | **Ratio[a]** | **Mean CvM statistic[b]** | **Mean ETS[c]** |
|---|---|---|---|---|
| Uncorrected humidity fields |  **90.9** |  **0.245** |  **134** |  **0.305** |
| Global humidity correction |  **91.1** |  **-0.014** |  **12.4** |  **0.319** |
| North Atlantic correction (Teoh et al., 2022) |  **90.7** |  **-0.104** |  **45.0** |  **0.324** |

[a]: Ratio compares the false positive and false negative rate and is computed by ($\frac{YN_{IAGOS}/NY_{HRES} \, (\%)}{NY_{IAGOS}/YN_{HRES} \, (\%)} - 1$). A positive value indicates that the ERA5 HRES underpredicts **ISSR occurrence**, a value of zero indicates a symmetrical false positive and false negative rate, while a negative value indicates that the ERA5 HRES overpredicts **ISSR occurrence**.

[b]: CvM test statistic, where a lower value indicates a better goodness-of-fit between the probability density function of the measured and ERA5-derived RHi.

[c]: The equitable threat score (ETS) is calculated according to Appendix A of Gierens et al. (2020), where ETS = 1 indicates that the ERA5-derived RHi is in perfect agreement with measurements, ETS = 0 indicates a completely random relationship, while ETS < 0 indicates an inverse relationship between the measured and ERA5-derived RHi.

  - [Supplement: Lines 226 – 241]: "When evaluated using four different performance metrics, the global humidity correction generally improved the agreement between RHi_IAGOS and RHi_corrected for each latitude bin (

Table S1). Table S**3**4 summarises the performance metrics when the full IAGOS dataset is compared with the uncorrected and corrected ERA5 HRES global humidity fields, showing **that**  the **weighted-mean**:

i. percentage of waypoints with the correct prediction of ISSR occurrence ($Y_{IAGOS}/Y_{HRES}$ and $N_{IAGOS}/N_{HRES}$) increased **by 0.2%** from **90.9**4% to **91.1**%,

ii. false positive ($N_{IAGOS}/Y_{HRES}$) and false negative ($Y_{IAGOS}/N_{HRES}$) rates are now symmetrical, meaning that errors in the ISSR occurrence and persistent contrail formation are expected to cancel out over the spatiotemporal domain,

iii. CvM test statistic reduced by 9**1**% (from **134** to **12.4**), which implies a significant improvement in the goodness-of-fit between the probability density function of $RHi_{IAGOS}$ and $RHi_{corrected}$ (Fig. S4), and

iv.  ETS improved **slightly** by 4.**4**% from 0.**305** to 0.313**, but the comparison at 0 – 40°N latitudes showed that the global humidity correction lowered the weighted-mean ETS by 3.9% from 0.281 and 0.270 (Table S1).**"

14. *Contrail formation and persistence will be highly sensitive to the vertical distribution of water vapour in the atmosphere. Given the authors have access to all the IAGOS data, it would be helpful and instructive to present more data on TP/FP/TN/FN rates (e.g., in the vertical and also by region) and to relate this to the challenges of doing a trajectory-based CoCiP analysis with all the associated specifics of aircraft type, engine emissions, etc.*

- Thank you for the detailed comment, we fully agree that further insights into the performance of the global humidity correction can be gained by segmenting the dataset not only by latitude, which was already presented in the Supplement, but also by altitude.

- To address this point, we have further segmented the IAGOS dataset by latitude and altitude which can be found in Table S4 of the revised Supplement,

    o [Supplement: Lines 206 – 220]:

**Table S4: Comparison of the ISSR occurrence between the in-situ RHi measurements from the IAGOS campaign in 2019 versus the RHi derived from the ERA5 HRES: (a) without humidity correction; and (b) with the global humidity correction, c.f. Eq. (S6) to Eq. (S9). The comparison is segmented into latitude intervals of 20° and altitude intervals of 4000 feet. $Y_{IAGOS}$ indicates that the waypoint has an RHi > 100% (ISSR occurrence) according to the IAGOS measurements, while $N_{IAGOS}$ indicates the opposite. The subscript "HRES" is used to indicate ISSR occurrence as provided by the ERA5 HRES. For (b), the metrics that are highlighted in green (red) indicates that its performance has improved (degraded) relative to (a).**

| | No. of waypoints | $Y_{IAGOS}/Y_{HRES}$ (%) | $N_{IAGOS}/N_{HRES}$ (%) | $Y_{IAGOS}/N_{HRES}$ (%) | $N_{IAGOS}/Y_{HRES}$ (%) | Ratio[a] | CvM stat[b] | ETS[c] |
|---|---|---|---|---|---|---|---|---|
| (a) | IAGOS vs. ERA5 HRES (No RHi correction) | | | | | | | |
| **0 - 20°N** | | | | | | | | |
| FL280-320 | 4615 | 5.26 | 81.0 | 8.21 | 5.53 | 0.486 | 12.5 | 0.217 |
| FL320-360 | 19580 | 6.38 | 79.5 | 7.56 | 6.56 | 0.152 | 75.7 | 0.245 |
| FL360-400 | 40529 | 6.60 | 78.5 | 5.96 | 8.98 | -0.336 | 78.5 | 0.237 |
| **20 - 40°N** | | | | | | | | |
| FL280-320 | 15989 | 4.32 | 86.1 | 4.98 | 4.58 | 0.087 | 12.3 | 0.267 |
| FL320-360 | 52857 | 5.08 | 86.6 | 4.97 | 3.31 | 0.502 | 104.0 | 0.338 |

| | | | | | | | | |
|---|---|---|---|---|---|---|---|---|
| FL360-400 | 184583 | 3.30 | 89.7 | 2.89 | 4.12 | -0.297 | 53.5 | 0.289 |
| **40 - 60°N** | | | | | | | | |
| FL280-320 | 13867 | 12.9 | 72.1 | 10.2 | 4.86 | 1.089 | 97.2 | 0.369 |
| FL320-360 | 79438 | 8.96 | 78.6 | 8.09 | 4.39 | 0.844 | 218.4 | 0.349 |
| FL360-400 | 122404 | 3.41 | 89.1 | 4.76 | 2.71 | 0.755 | 172.4 | 0.280 |
| **(b)  IAGOS vs. ERA5 HRES (Global humidity correction)** | | | | | | | | |
| **0 - 20°N** | | | | | | | | |
| FL280-320 | 4615 | 4.46 | 81.6 | 9.01 | 4.88 | 0.849 | 5.78 | 0.187 |
| FL320-360 | 19580 | 5.32 | 80.5 | 8.62 | 5.55 | 0.552 | 26.2 | 0.212 |
| FL360-400 | 40529 | 5.90 | 79.6 | 6.66 | 7.80 | -0.146 | 10.6 | 0.224 |
| **20 - 40°N** | | | | | | | | |
| FL280-320 | 15989 | 3.88 | 86.9 | 5.41 | 3.85 | 0.404 | 4.90 | 0.255 |
| FL320-360 | 52857 | 4.70 | 87.0 | 5.36 | 2.94 | 0.824 | 63.5 | 0.321 |
| FL360-400 | 184583 | 3.11 | 90.1 | 3.08 | 3.72 | -0.171 | 7.79 | 0.283 |
| **40 - 60°N** | | | | | | | | |
| FL280-320 | 13867 | 15.2 | 70.3 | 7.82 | 6.71 | 0.166 | 5.73 | 0.412 |
| FL320-360 | 79438 | 11.1 | 76.6 | 5.98 | 6.33 | -0.056 | 14.4 | 0.397 |
| FL360-400 | 122404 | 4.36 | 87.9 | 3.81 | 3.91 | -0.025 | 2.06 | 0.323 |

[a]: Ratio compares the false positive and false negative rate and is computed by $(\frac{Y_{IAGOS}/N_{HRES}\,(\%)}{N_{IAGOS}/Y_{HRES}\,(\%)} - 1)$. A positive value indicates that the ERA5 HRES underpredicts ISSR occurrence, a value of zero indicates a symmetrical false positive and false negative rate, while a negative value indicates that the ERA5 HRES overpredicts ISSR occurrence.

[b]: CvM test statistic, where a lower value indicates a better goodness-of-fit between the probability density function of the measured and ERA5-derived RHi.

[c]: The equitable threat score (ETS) is calculated according to Appendix A of Gierens et al. (2020), where ETS = 1 indicates that the ERA5-derived RHi is in perfect agreement with measurements, ETS = 0 indicates a completely random relationship, while ETS < 0 indicates an inverse relationship between the measured and ERA5-derived RHi.

- We have revised the supplement to include a short discussion of the IAGOS dataset when it is segmented by latitude and altitude. Based on these results, we also highlight the potential limitations of the global humidity correction model:

    o [Supplement: Lines 242 – 256] "**To evaluate the agreement between RHi$_{IAGOS}$ and RHi$_{corrected}$ at different altitudes, we also segmented the full IAGOS dataset by latitude intervals of 20° and altitude intervals of 4000 feet (Table S4). The results showed that:**

    i. **the CvM test statistic improved across every latitude and altitude categories, suggesting an improved goodness-of-fit between the probability density function of RHi$_{IAGOS}$ and RHi$_{corrected}$, but**

    ii. **weighted-mean ETS degraded from 0.286 to 0.275 (-3.8%) across all altitude intervals at lower latitudes (0 – 40°N), and**

    iii. **the weighted-mean ratio of false negative-to-false positive rate $(\frac{Y_{IAGOS}/N_{HRES}\,(\%)}{N_{IAGOS}/Y_{HRES}\,(\%)} - 1)$ at lower latitudes (0 – 40°N) and altitudes (28,000 – 36,000 feet) increased by around two-fold from 0.356 (no humidity correction) to 0.670 (with global humidity corrections applied). In other words, the uncorrected humidity fields are already underestimating the ISSR occurrence in this region (0 – 40°N and 28,000 – 36,000 feet); and applying the global humidity correction could potentially worsen this underestimation.**

    **Points (ii) and (iii) suggest that there could be potential limitations in the global humidity correction when applied at lower latitudes (0 – 40°N) and altitudes (28,000 – 36,000 feet), where the air traffic activity in this region**

**(0 – 40°N and 28,000 – 36,000 feet) accounted for 20.9% of the global annual flight distance flown in 2019. However, given the limited sample size within this region (0 – 40°N and 28,000 – 36,000 feet, which collectively constitute only 14% of the full IAGOS dataset), we have opted against any attempts to rectify these biases.**"

- In addition, we have also conducted an additional analysis to confirm that the lower relative contrail occurrence in the subtropics (as shown in Fig. 5 in the revised manuscript) is not an artefact of the global humidity correction:

    o [Supplement: Lines 470 – 472] "**A comparison of the $p_{contrail}$ by latitude for the simulation with and without humidity correction (Fig. S18) confirms that the minimum pcontrail observed at the subtropics (30°N/S ± 5°) is not an artefact of the global humidity correction.**"

    o [Supplement: Lines 481 – 484]

[Figure]

**Figure S3: The percentage of annual flight distance flown that formed persistent contrails ($p_{contrail}$) in 2019 in the simulation with (blue line) and without (orange line) global humidity corrections applied to the ERA5 HRES (blue line).**

15. *How does water vapour uncertainty propagate into a calculation of radiative impact at the global scale? There are no errors presented on annual net RF values. What are these errors and how much is attributed to water vapour uncertainty?*

- Thank you for this suggestion. In our manuscript, we conducted a sensitivity analysis to evaluate the relative importance of various input parameters and contrail model assumptions to the 2019 global annual mean contrail net radiative forcing (RF). Although we do not directly quantify the uncertainties attributed to water vapour uncertainty in this analysis, we found that the RF estimates were most sensitive to the global humidity correction, followed by the aircraft particle number emissions, and is least sensitive to contrail model assumptions (i.e., radiative heating and contrail-contrail overlapping).

- To estimate the uncertainty of the global annual mean net radiative forcing (RF) resulting from meteorological uncertainties require the use of a Monte Carlo simulation. While we acknowledge the importance of this aspect, conducting such simulations is computationally intensive and falls outside the scope of this study. We

note that this specific research question is currently being addressed in a separate manuscript, which has since been submitted for peer review.

**ADDITIONAL CORRECTIONS**

16. We have corrected a typographical error in Eq. (2) in the manuscript, where one of the coefficients in the denominator has been updated from 0.04589 to the correct value of 0.4589.

17. We have corrected a typographical error in Fig. 9c and Fig. 9d in the revised manuscript, where the x-axis should be labelled as the "latitude" instead of "longitude".

18. We have replaced the term "overall propulsion efficiency" with "overall efficiency" to avoid confusion between the terms "propulsion efficiency" and "overall efficiency".

**REFERENCES**

[revised manuscript text omitted]